# OmniVideoBench: Towards Audio-Visual Understanding Evaluation for Omni MLLMs

Caorui Li[1,2*], Yu Chen[1,2*], Yiyan Ji[1*],
Jin Xu[3], Zhenyu Cui[2], Shihao Li[1], Yuanxing Zhang[4], Zhenghao Song[5],
Dingling Zhang[1], Ying He[6], Haoxiang Liu[6], Yuxuan Wang[3], Qiufeng Wang[2],
Jiafu Tang[1], Zhenhe Wu[5], Jiehui Luo[7], Zhiyu Pan[1], Weihao Xie[8], Chenchen Zhang[5],
Zhaohui Wang[1], Jiayi Tian[3], Yanghai Wang[1], Zhe Cao[1], Minxin Dai[1], Ke Wang[5],
Runzhe Wen[1], Yinghao Ma[9], Yaning Pan[10], Sungkyun Chang[9], Termeh Taheri[9],
Haiwen Xia[11], Christos Plachouras[9], Emmanouil Benetos[9], Yizhi Li[13],
Ge Zhang[5], Jian Yang[5], Tianhao Peng[5], Zili Wang[5], Minghao Liu[12],
Junran Peng[6], Zhaoxiang Zhang[14], Jiaheng Liu[1†]

[1]Nanjing University    [2]Southeast University    [3]Alibaba Group    [4]Kuaishou Technology
[5]M-A-P    [6]University of Science and Technology Beijing    [7]Central Conservatory of Music
[8]Huazhong University of Science and Technology    [9]Queen Mary University of London
[10]Fudan University    [11]Peking University    [12]2077AI
[13]University of Manchester    [14]Chinese Academy of Sciences
caoruili@seu.edu.cn
liujiaheng@nju.edu.cn

## Abstract

Recent advances in multimodal large language models (MLLMs) have demonstrated substantial potential in video understanding. However, existing benchmarks fail to comprehensively evaluate synergistic reasoning capabilities across audio and visual modalities, often neglecting either one of the modalities or integrating them in a logically inconsistent manner. To bridge this gap, we introduce **OmniVideoBench**[1], a large-scale and rigorously designed benchmark dedicated to assessing synergistic audio–visual understanding, with a strong emphasis on modality complementarity and logical consistency. Specifically, OmniVideoBench comprises **1000** high-quality question-answer(QA) pairs, each annotated with step-by-step reasoning traces, derived from 628 diverse videos ranging from **several seconds to 30 minutes**, and manually verified to guarantee complete correctness and uniqueness. Moreover, OmniVideoBench encompasses **13** carefully designed question types, covering temporal reasoning, spatial localization, counting, causal inference, summarization, and beyond, thereby capturing the essential challenges of video understanding. Evaluation of multiple MLLMs on OmniVideoBench reveals a pronounced gap between model performance and human reasoning, with open-source models lagging significantly behind their closed-source counterparts, underscoring the inherent difficulty of genuine audio–visual reasoning. We will release OmniVideoBench to foster the development of MLLMs with stronger and more generalizable reasoning capabilities.

## 1 Introduction

Multimodal large language models (MLLMs) have recently made impressive progress in bridging vision, language, and audio (Yin et al., 2024; Song et al., 2025; Cheng et al., 2025). While early benchmarks primarily focused on image–text alignment or visual reasoning (Xu et al., 2025c; Chen et al., 2024c; Yue et al., 2024a), the integration of video and audio presents a quite different challenge: models must jointly process long temporal sequences, dynamic scene transitions, and com-

---

* Equal contribution. This work was conducted during the author's internship at Nanjing University.
† Corresponding Author.
[1]https://github.com/NJU-LINK/OmniVideoBench

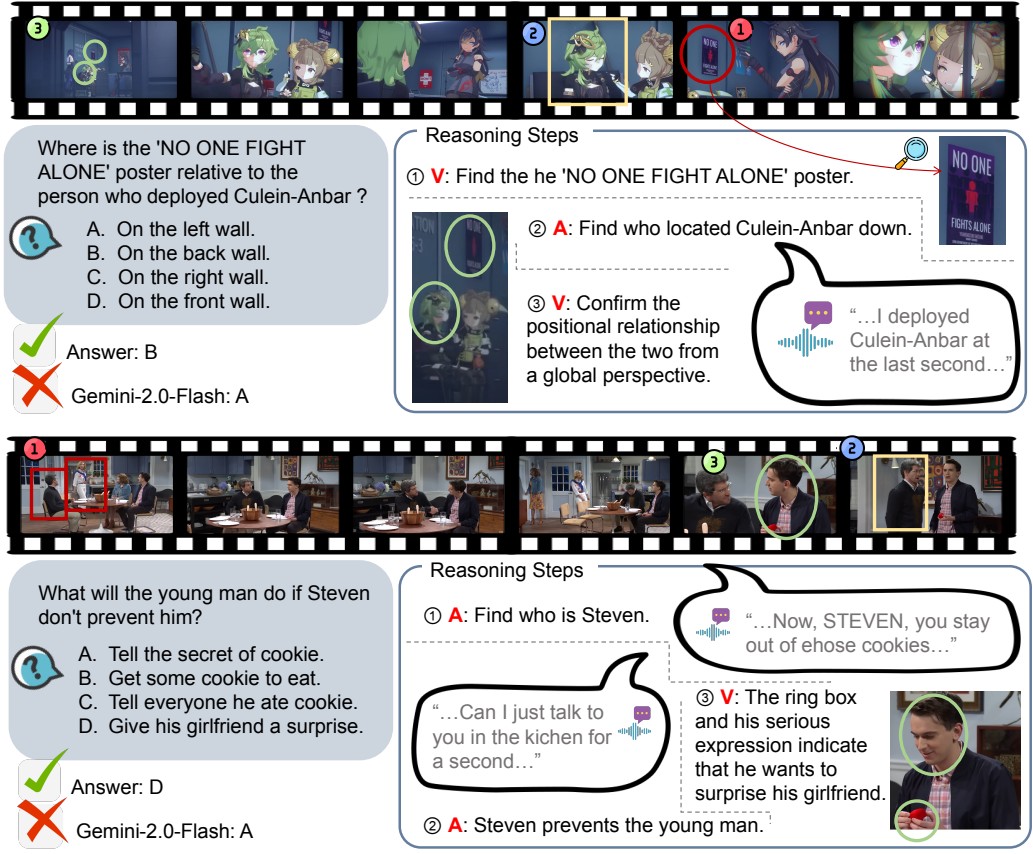

Figure 1: Examples in OmniVideoBench ("V" presents vision and "A" presents audio), and we present the atomic reasoning traces for these examples.

plementary acoustic cues. Despite rapid advances, evaluation of MLLMs on audio–visual reasoning remains underdeveloped. Existing benchmarks (Li et al., 2024a; Hong et al., 2025) often *(i) focus on **short video clips** that underrepresent long-term temporal dependencies, (ii) emphasize a **single modality** (e.g., vision) while treating audio as auxiliary or optional.* As a result, current evaluations fail to capture the challenges inherent to comprehensive video understanding, where audio and vision must be integrated consistently and logically to support robust inference.

To address these limitations, we introduce **OmniVideoBench**, a high-quality benchmark designed for evaluating audio–visual reasoning abilities in MLLMs. Specifically, first, we collect 628 diverse videos spanning up to 30 minutes across 8 major categories and 68 subcategories, covering realistic contexts such as news, sports, documentaries, vlogs, and ego-centric recordings. Then, we construct 1,000 high-quality question–answer pairs based on these videos, and each pair is annotated with step-by-step reasoning chains as shown in Figure 1, where these reasoning steps explicitly indicate modality and evidence information. This design not only strengthens the reliability of the evaluation but also provides a unique signal for analyzing how models reason, rather than just the final answers.

Based on our OmniVideoBench, we conduct extensive evaluations of both closed-source and open-source MLLMs, and several insightful findings are as follows:

- **OmniVideoBench poses significant challenges for Omni-Modal Language Models.** Current MLLMs have not achieved a passing score (<60%) on OmniVideoBench. The best-performing model, Gemini-2.0-Pro, only achieves an accuracy of 58.90%. Except for the newly proposed Qwen3-Omni, the performance of open-source models is close to random.

- **Omni-understanding abilities on long videos have significant improvement room.** Although some leading models (such as Gemini-2.5-pro) demonstrate relatively robust perfor-

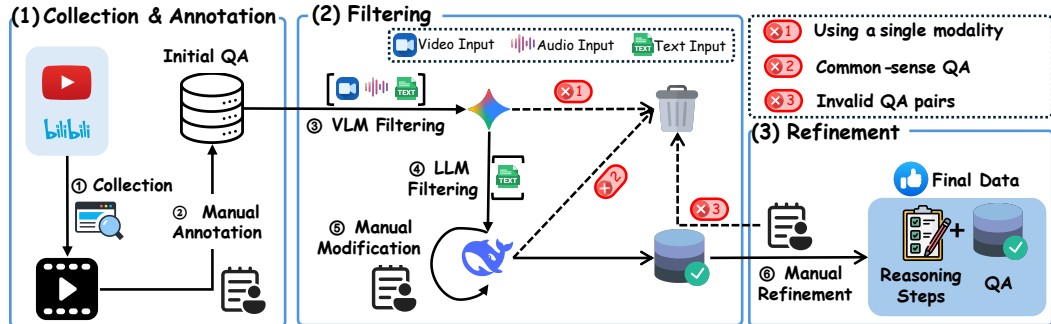

Figure 2: The complete pipeline of data collection, annotation, and refinement, where filtering and refinement serve as two key processes for quality assurance.

mance on long videos, other models (e.g., Gemini-2.0-Flash, Qwen3-Omni-30B-A3B) still struggle on long video understanding.

- **Performance varies a lot for videos with different audio signals.** Gemini-2.5-Pro only achieves 38.46% accuracy on videos with music signal, while the results on sound and speech are 57.72% and 61.66%, respectively.

- **Performance on different task types differs a lot.** For example, Gemini-2.5-Pro achieves accuracy below 50% on the background and music understanding task, which requires low-semantic acoustic cues (e.g., musical style, tempo changes), and the accuracy results on the relationship reasoning and summarization tasks are more than 80%.

## 2 OMNIVIDEOBENCH

### 2.1 OVERVIEW

OmniVideoBench is a benchmark for evaluating the audio-visual collaborative reasoning of MLLMs. The main task in the evaluation requires a model to process a video, its audio, and associated text to generate a textual answer supported by explicit reasoning steps. This process assesses the model's ability to synthesize information across modalities, from recognizing objects to comprehending complex scene dynamics and context. This section details the benchmark's design principles, annotation protocols, and dataset statistics.

### 2.2 VIDEO COLLECTION

OmniVideoBench is composed of real-world videos sourced from YouTube[2] and Bilibili[3]. These videos feature rich audiovisual content; therefore, comprehensive understanding necessitates the accurate processing and integration of both audio and visual modalities for reasoning.

Regarding video richness, we primarily focus on two dimensions: type and duration. For type diversity, we categorize videos into eight broad classes: Vlog, News, Cartoon, Sports, Documentary, TV, Ego, and Others. Each class is further subdivided into nearly seventy fine-grained subcategories, which facilitates video retrieval and ensures broad coverage. Video categories are unevenly distributed. News and documentary videos have dense audio that nearly covers visual content, making them unsuitable for audio-visual reasoning tasks; thus, we manually controlled the video type distribution. For duration diversity, we restrict video lengths to the range of several seconds to 30 minutes, so as to evaluate reasoning across varying temporal scales.

Building upon this foundation, we established a set of rigorous video collection criteria that not only ensure the quality of the videos themselves, like resolution, but also guarantee the richness and diversity of their audio and visual content. To further avoid data overlap with existing training sets

---

[2]https://www.youtube.com/
[3]https://www.bilibili.com/

(e.g., popular TV shows), we restrict the selection to recent publications. The detailed collection principles are provided in Appendix B.

## 2.3 DATA ANNOTATION

After collecting high-quality videos, we carried out manual annotation. Compared with automated annotation, automated methods cap the evaluation ceiling by the capabilities of the annotating model, whereas manual annotation produces questions that are closer to real-world needs.

In Figure 2, we first designed multiple-choice questions consisting of the question stem, the correct answer, and several distractors, to facilitate convenient evaluation of model performance. At this stage, we obtained approximately 2,500 QA pairs. We categorize the tasks into 13 types: Fine-grained Perception, Spatial Reasoning, Attribute Comparison, Background & Music Understanding, Counting, Temporal Understanding, Summarization, Sentiment Analysis, Causal Reasoning, Relationship Reasoning, Reference Reasoning, Ego Reasoning, and Hypothetical Reasoning. In this design, each question is required to rely on audio-visual reasoning, and the answer must be both correct and unique with no alternative plausible interpretations in the video. Moreover, we require that questions should not depend on video resolution or frame rate. Cases where the target object is extremely small, blurred, and barely recognizable to the human eye, or where the relevant event occurs only within an instant, are excluded.In addition, we established the following rules to minimize the interference caused by extraneous textual information.

- **Questions should avoid redundant information.** We minimize unnecessary details in the question text, such as the gender, clothing, or exact speech of characters, as long as doing so does not affect the correctness or uniqueness of the answer. This serves two purposes: reducing textual cues the model could exploit and increasing question difficulty to better test its audio-visual understanding.

- **The length of answers is capped.** To prevent the answer text itself from providing excessive cues to the model, which could reduce the extent to which the evaluation reflects its understanding and reasoning over audio and visual modalities, we impose a limit on answer length. This constraint ensures that the results more faithfully capture the model's multimodal comprehension and reasoning capabilities.

- **The format of options must be consistent.** Here, "format" refers to aspects such as length, tone, style, and variation patterns. If these features are inconsistent, they may provide the model with unintended cues for reasoning. For instance, when three options are considerably longer than the remaining one, when three options adopt a casual tone while the other is markedly formal, such discrepancies undermine the assumption that each option should have an equal probability of being chosen, thereby compromising the fairness of the evaluation.

- **Negative options must be relevant to the question.** We require that all distractors appear in the video and maintain relevance to the question. Without this constraint, the model could easily eliminate distrastors, greatly reducing the need for reasoning.

- **Options should maintain a consistent semantic distance.** We formalize semantic distance as the number of differing semantic units between options. Let an option $o_i$ be represented as a set of semantic units $S_i$. The semantic distance between two options $o_i$ and $o_j$ is defined as:

$$d(o_i, o_j) = |S_i \triangle S_j| \tag{1}$$

  where $\triangle$ denotes the symmetric difference, capturing the distinct semantic units between two options. To prevent models from exploiting unbalanced textual cues rather than performing genuine audio-visual reasoning, we require that all distractors have consistent distances from one another and from the correct option.

## 2.4 QUALITY ASSURANCE

We employed an advanced MLLM (i.e., Gemini 2.0 Flash), with strong audiovisual perception and comprehension capabilities, as well as long-context processing ability, to filter out questions that could be resolved using only a single modality. If the model successfully selected the correct answer with a plausible explanation while relying solely on unimodal information, the corresponding question was removed. After this filtering stage, approximately 1,500 questions were retained.

Table 1: Dataset statistics divided into video-level and annotation-level information.

| Video Statistics | | Annotation Statistics | |
|---|---|---|---|
| #Major Categories | 8 | #Task Types | 13 |
| #Subcategories | 68 | Avg. Question Len. | 14.68 words |
| Avg. Duration | 384.24 s | Avg. Answer Len. | 4.92 words |
| Min. Resolution | 480p | Avg. Reasoning Steps | 5.68 |
| Max. Resolution | 1080p | Audio Types (Sp:So:Mu) | 762:147:91 |

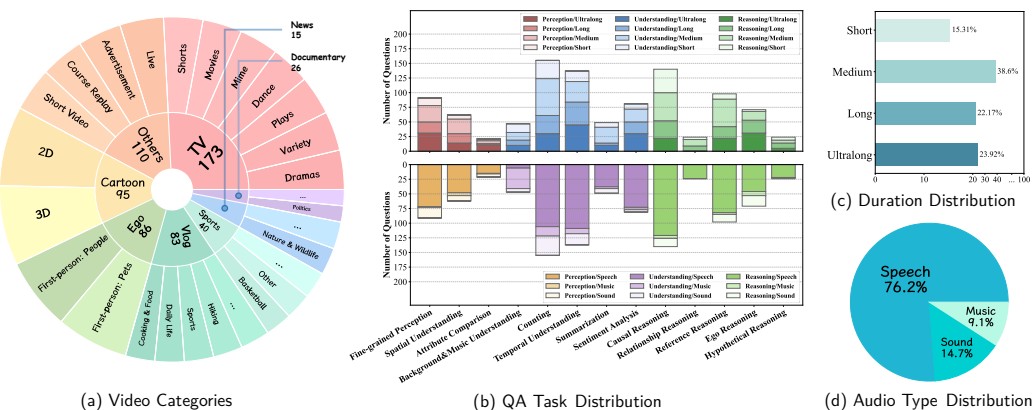

(a) Video Categories  (b) QA Task Distribution  (c) Audio Type Distribution

Figure 3: (a) OmniVideoBench covers 8 major categories and 68 subcategories. (b) OmniVideoBench comprises 13 task types. The above part shows the video duration distribution across different tasks, while the durations are categorized into four groups: **"Short"** for less than 1 minute, **"Medium"** for 1–5 minutes, **"Long"** for 5–10 minutes, and **"Ultralong"** for more than 10 minutes. The lower part illustrates the distribution of three types of audio (i.e., **Speech**, **Sound** and **Music**). (c) Distribution of video durations across four time intervals. (d) Distribution of three audio types.

Subsequently, we employed a large language model, DeepSeek-V3.1 (Liu et al., 2024a), with strong reasoning capabilities to filter out questions that could be answered solely based on textual information. Such cases primarily fall into two categories: first, questions that involve classical, well-known, or universally shared knowledge or objects, which can be answered without reference to the video content; and second, questions where the phrasing of the question, options, or answers provides unintended textual cues. For the former, we directly discarded the questions. For the latter, our annotators reviewed the reasoning process generated by the model and revised the textual formulations to eliminate such biases. After this stage of refinement, 1103 questions were retained.

Another group of annotators conducts the final refinement stage, thoroughly reviewing all questions to identify and remove those with incorrect, non-unique, or mismatched answers. After this validation, annotators enriched each question with step-by-step reasoning chains, where each step consists of three elements: modality, evidence, and inference. The modality specifies whether the step relies on audio or visual information; the evidence denotes the specific information extracted from the video; and the inference describes the reasoning derived from that information. We required each step to be atomic, meaning that it should involve only one modality and capture a minimal unit of evidence, such as a spoken sentence, an action, or the appearance of a character. This design ensures that the reasoning process is both detailed and comprehensive. Through this process, we obtained 1000 high-quality QA pairs with explicit step-by-step reasoning chains, forming a robust dataset for multimodal audio-visual reasoning.

## 2.5 DATASET STATISTICS

As shown in Table 1, our OmniVideoBench dataset consists of 628 real-world videos with audio tracks, spanning 8 major categories and 68 subcategories. The videos are of high quality and diverse in content, with an average duration of 384.6 seconds, an average resolution of 480p, about 2k ASR-transcribed tokens per video, and roughly three speakers per video. On the annotation side, OmniVideoBench contains 1000 audio–visual reasoning QA pairs across 13 task types, with an

| Benchmark | Modality | Qwen2.5-Omni | Multiple Domains | Video Type | Audio Type | Video Duration | Answer Type |
|---|---|---|---|---|---|---|---|
| AVQA (Yang et al., 2022) | V+A | / | ✗ | R | So | 10 | MC |
| Music-AVQA (Li et al., 2022) | V+A | / | ✗ | R+S | Mu | 60 | CLS |
| AVTRUSTBENCH (Chowdhury et al., 2025) | V+A | / | ✓ | R+S | Sp+So+Mu | 10\60 | MC |
| MMAU (Sakshi et al., 2024) | A | 71.0 | ✓ | / | Sp+So+Mu | / | MC |
| DAVE (Radevski et al., 2025) | V+A | 31.0 | ✓ | R+S | So | ≤ 60 | MC |
| AV-Odyssey (Gong et al., 2024) | I+A | / | ✓ | R | Sp+So+Mu | / | MC |
| AVHBench (Judgement) (Sung-Bin et al., 2024) | V+A | 74.7 | ✓ | R+S | So | 10 | CLS |
| OmniBench (Li et al., 2024c) | I+A | 56.1 | ✓ | R | Sp+So+Mu | / | MC |
| Daily-Omni (Zhou et al., 2025) | V+A | 47.5 | ✗ | R | Sp+So+Mu | 30\60 | MC |
| WorldSense (Hong et al., 2025) | V+A | 48.3 | ✓ | R | Sp+So+Mu | 15–656 | MC |
| **OmniVideoBench (Ours)** | V+A | 29.3 | ✓ | R | Sp+So+Mu | 4–1955 | MC |

Table 2: **Comparisons between different benchmarks and datasets.** V, I, A for modality represent video, image and audio. **Qwen2.5-Omni** represents the performance of Qwen2.5-Omni-7B on these benchmarks. **Multiple Domains** signifies whether the video includes diverse domains. R and S in **Video Type** denote real-world and synthetic data. Sp, So, and Mu represent Speech, Sound, and Music for **Audio Type**, respectively. **Video Duration** represents the duration in seconds. MC, CLS for **Answer Type** indicate Multiple Choice and Classification from fixed vocabulary, respectively.

average question length of 14.68 words and an average answer length of 4.92 words. Each QA pair is annotated with step-by-step reasoning chains averaging 5.68 steps. The reasoning process covers both modalities, with 54% of steps grounded in vision and 46% in audio. There are 762, 147, 91 QA pairs related to Speech, Sound and Music, respectively, highlighting the complementarity of modalities in multi-step reasoning. Moreover, we provide more detailed statistics in Figure 3.

## 2.6 DATASET COMPARISON

As shown in Table 2, we compare OmniVideoBench with representative audio-video benchmarks. While AV-Odyssey (Gong et al., 2024) and OmniBench (Li et al., 2024c) operate on single images, OmniVideoBench targets substantially more challenging videos with durations ranging from a few seconds to 30 minutes. Recent benchmarks (Chowdhury et al., 2025; Radevski et al., 2025; Sakshi et al., 2024; Sung-Bin et al., 2024) have begun to emphasize audio–video coordination; however, they typically focus on specific capabilities or short clips. Some also evaluate audiovisual consistency, yet their tasks remain primarily centered on shorter videos and hallucination detection. OmniVideoBench, by contrast, expands the scope to diverse video types, broader temporal spans, and fine-grained cross-modal reasoning, capturing richer dependencies between audio and vision. Some other audio-visual benchmarks (Zhou et al., 2025; Hong et al.,

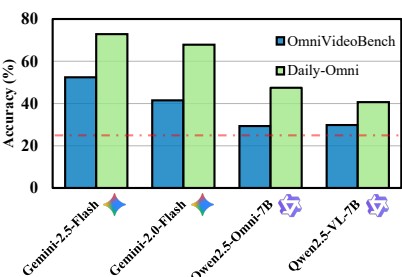

Figure 4: Performance comparison of selected models on OmniVideoBench and Daily-Omni. "Red line" denotes random guessing.

2025), do not fully achieve a natural integration of the two modalities, while OmniVideoBench addresses this limitation effectively. For instance, disabling audio causes Gemini-2.0-Flash's performance to plummet to the near-random level, indicating that visual-only cues are insufficient. Furthermore, Figure 4 shows that widely used models such as Qwen2.5-Omni-7B perform closer to random guessing on our benchmark, indicating that OmniVideoBench presents significantly greater challenges than existing multimodal datasets.

## 3 EXPERIMENTS

### 3.1 BASELINE MODELS

We evaluate open-source MLLMs (i.e., Qwen3-Omni series (Xu et al., 2025b), Qwen2.5-Omni series (Xu et al., 2025a), Baichuan-Omni-1.5 (Li et al., 2025), HumanOmni (Zhao et al., 2025), MiniCPM-o (Yao et al., 2024), VideoLLaMA2 (Cheng et al., 2024)), VITA-1.5-7B (Fu et al., 2025),

| Models | Audio Type | | | Video Duration | | | | Avg. |
|---|---|---|---|---|---|---|---|---|
| | Music | Sound | Speech | (0,1] min | (1,5] min | (5,10] min | (10,30] min | |
| *Omni-Modal Language Models (With Visual and Audio)* | | | | | | | | |
| Gemini-3.0-Pro | **52.81** | 55.17 | **64.13** | **62.42** | **66.18** | **57.02** | **59.76** | **61.80** |
| Gemini-2.5-Pro | 38.46 | **57.72** | 61.66 | 57.83 | 64.43 | 55.02 | 55.94 | 58.90 |
| Gemini-3.0-Flash | 49.45 | 50.34 | 56.69 | 58.43 | 55.10 | 55.90 | 52.29 | 55.10 |
| Gemini-2.5-Flash | 39.56 | 57.04 | 53.17 | 55.42 | 55.10 | 47.37 | 52.11 | 52.40 |
| Gemini-2.0-Flash | 29.67 | 40.27 | 43.21 | 49.40 | 43.15 | 41.05 | 34.87 | 41.50 |
| Qwen3-Omni-30B-A3B | 37.36 | 34.67 | 39.26 | 45.78 | 37.03 | 38.86 | 35.11 | 38.40 |
| OmniVinci-9B | 30.77 | 32.67 | 32.15 | 38.55 | 34.11 | 30.13 | 27.10 | 32.10 |
| Baichuan-Omni-1.5 | 24.18 | 31.33 | 31.36 | 28.92 | 31.78 | 28.38 | 32.44 | 30.70 |
| HumanOmni-7B | 20.87 | 31.08 | 31.61 | 36.57 | 29.36 | 29.60 | 29.25 | 30.50 |
| VITA-1.5-7B | 25.27 | 28.57 | 31.49 | 31.33 | 27.41 | 30.57 | 33.97 | 30.50 |
| MiniCPM-o | 27.47 | 28.57 | 30.24 | 31.43 | 28.49 | 34.53 | 26.15 | 29.70 |
| Qwen2.5-Omni-7B | 23.07 | 25.33 | 30.70 | 41.57 | 27.41 | 25.33 | 26.72 | 29.30 |
| VideoLLaMA2-7B | 26.37 | 30.67 | 29.25 | 32.00 | 28.20 | 29.60 | 28.29 | 29.20 |
| *Omni-Modal Language Models (Visual Only)* | | | | | | | | |
| Gemini-2.0-Flash | 25.27 | 36.67 | 30.99 | 33.73 | 35.86 | 32.75 | 22.48 | 31.30 |
| Qwen2.5-Omni-7B | 27.47 | 26.67 | 26.22 | 28.31 | 27.11 | 24.45 | 25.95 | 26.40 |
| *Visual Language Models (Visual Only)* | | | | | | | | |
| Qwen2.5-VL-32B | 32.97 | 32.00 | 31.49 | 38.55 | 31.20 | 29.26 | 30.53 | 31.80 |
| Qwen2.5-VL-7B | 29.67 | 31.33 | 29.51 | 25.90 | 30.03 | 31.88 | 30.15 | 29.80 |
| Qwen2.5-VL-72B | 26.37 | 29.33 | 29.91 | 33.13 | 30.03 | 31.88 | 24.43 | 29.50 |
| *Baseline LLMs* | | | | | | | | |
| DeepSeek-V3.1 | 28.57 | 26.17 | 27.28 | 30.91 | 27.57 | 25.00 | 26.44 | 27.60 |

Table 3: Results of different models. The table reports accuracy on videos across three audio types and four duration ranges. Boldface highlights the best performance within each column.

OmniVinci-9B (Ye et al., 2025) and various closed-source MLLMs (i.e., Gemini-2.5-Pro, Gemini-2.5-Flash (Comanici et al., 2025), and Gemini-2.0-Flash). We also evaluate the Qwen2.5-VL series (Bai et al., 2025) and DeepSeek-V3.1 (Liu et al., 2024a).

## 3.2 HUMAN PERFORMANCE

We invited 10 qualified annotators, including 8 graduate students experienced in multimodal research and 2 experts trained in music-related analysis. Before the evaluation, 50 questions were randomly selected for 10 testers to answer simultaneously. The fact that the difference in results was no more than 5 questions indicates that the manual evaluation has minimal deviation and is viable. We consolidated questions requiring musical knowledge and assigned them to two music experts. The remaining questions were divided to ensure roughly equal distribution of speech and sound audio samples per set, which were then evenly assigned to high-level personnel such as graduate students to obtain persuasive results. Manual responses had no time constraints, yielding a final accuracy rate of **82.69%**. This demonstrates that existing models still fall significantly short of human capabilities.

## 3.3 MAIN RESULTS

In Table 3, we present evaluation results on OmniVideoBench and have the following observations:

- **Open-source models still lag significantly behind closed-source models.** Gemini-2.5-Pro achieves the best performance across most tasks. This underscores the urgent need for current open-source models to improve in multiple areas, including fine-grained perception, cross-modal reasoning, and speech awareness.

- **MLLMs show a performance degradation when dealing with music-related audio.** We observe that models exhibit lower accuracy in responding to music-dominated videos compared to those containing human voices or ambient sounds, a phenomenon particularly pronounced in open-source models. Unlike human voices conveying explicit semantic content or ambient sounds often corresponding to specific visual events, music primarily encodes abstract

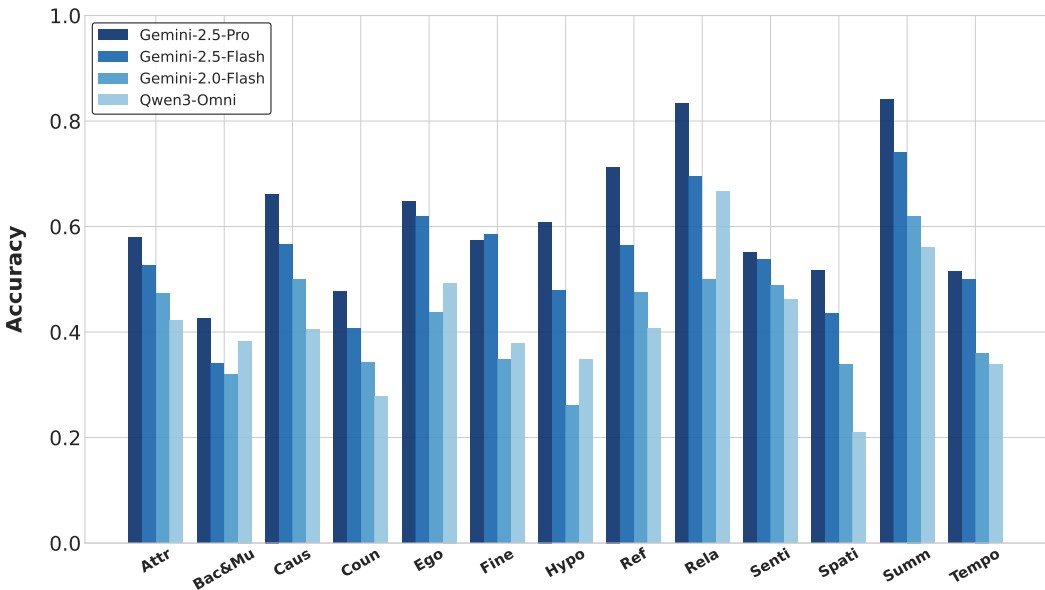

Figure 5: Performance Comparison of some Open-Source and Closed-Source Omni Models on 13 Tasks in OmniVideoBench. Here, "**Attr**": Attribute Comparison, "**Bac&Mu**": Background and Music Understanding, "**Caus**": Cause and Effect Reasoning, "**Coun**": Counting, "**Ego**": Ego Reasoning, "**Fine**": Fine-grained Perception, "**Hypo**": Hypothetical Reasoning, "**Ref**": Referential Reasoning, "**Rela**": Relationship Reasoning, "**Senti**": Sentiment Analysis, "**Spati**": Spatial Reasoning, "**Summ**": Summarization, "**Tempo**": Temporal Sequencing Understanding.

emotional and atmospheric information. Current MLLMs demonstrate limited capability to translate such implicit cues into effective reasoning, indicating that cross-modal alignment for emotional and atmospheric understanding remains an urgent challenge to be addressed.

- **Current MLLMs still have room for improvement in long videos.** Although some leading models like Gemini-2.5-Pro demonstrate relatively robust performance on long videos, most MLLMs (e.g., Gemini-2.0-Flash, Qwen3-Omni) still struggle in long videos, which highlights the widespread challenge in understanding long videos.

## 3.4 FURTHER ANALYSIS

**Performance of Models on Tasks across Different Types.** Figure 5 presents a fine-grained comparison of model accuracy on the 13 reasoning categories in OmniVideoBench. Several consistent patterns emerge. (1). Closed-source MLLMs demonstrate superior performance across nearly all task types. Gemini-2.5-Pro achieves the highest accuracy on 11 out of 13 tasks, demonstrating particularly strong performance in *Relationship Reasoning*, *Spatial Reasoning*, *Referential Reasoning*, and *Cause and Effect Reasoning*. These tasks require long-term sequence integration and multi-step cross-modal reasoning, highlighting Gemini's strengths in long-context modeling and multimodal fusion. (2). MLLMs' understanding of audio remains limited to relatively superficial surface-level information. Whether open-source or closed-source models, *Background and Music Understanding* remains the most challenging task, with even Gemini-2.5-Pro achieving accuracy below 50%. This is probably because such tasks require linking low-semantic acoustic cues (e.g., musical style, tempo changes) with high-level reasoning, while current models struggle to master the capability. In contrast, *Relationship Reasoning* and *Summarization* are relatively easier. This may be because they rely more on recognizing language within audio and visual observation capabilities, and less on cross-modal abstraction abilities.

**Effect of ASR Transcripts for Visual Only MLLMs.** To further investigate the role of audio information in MLLMs' reasoning performance, we evaluate several models using both the automatic speech recognition (ASR) transcripts generated by the Voxtral-Mini-3B model (Liu et al., 2025a)

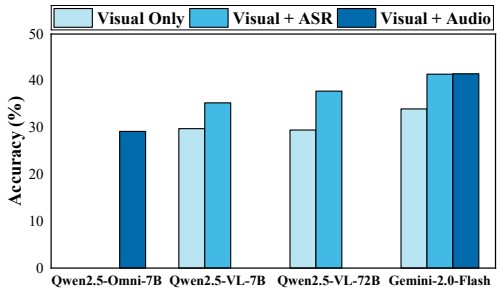

(a) Accuracy rates of selected MLLMs under different inputs.

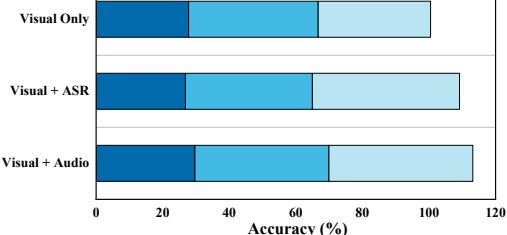

(b) Accuracy of Gemini-2.0-Flash on videos with different audio types.

Figure 6: Accuracy comparison of MLLMs with and without ASR transcripts on OmniVideoBench.

and silent video frames as inputs. The results are shown in Figure 6. The observations are as follows: (1). Open-source models demonstrate weaker integration capabilities for audio information compared to their understanding of textual information. In Figure 6a, all tested models demonstrate significantly improved accuracy after extracting ASR text information compared to receiving only visual inputs. However, the Qwen2.5-Omni-7B model, which processes both visual and audio inputs simultaneously, performed even worse than the Qwen2.5-VL-7B model with equivalent parameters. This highlights a common challenge faced by most open-source Omni-Modal Language Models: insufficient cross-modal reasoning capabilities for audio-visual information. (2). In cross-modal video reasoning, audio comprehension capabilities remain irreplaceable by ASR. In Figure 6b, although ASR can help MLLMs achieve decent performance on certain tasks requiring speech recognition capabilities, its effectiveness is extremely limited for tasks demanding deeper and more abstract audio comprehension such as the videos whose audio type is *Music* or *Sound*.

**Effect of Different Numbers of Frames.** We conduct experiments on Qwen2.5-Omni-7B and Qwen3-Omni-30B-A3B with total frame counts fixed at 32, 64, 128, and 256, respectively, and observe that both models benefit from more frequent time sampling. In Figure 7a, as the total frame counts increase, accuracy steadily improves, likely because richer temporal coverage provides more complete motion cues and reduces the risk of missing key events. As shown in Figure 7b, this improvement becomes more pronounced for longer videos. The consistent gains across different video durations further indicate that dense frame sampling not only captures fine-grained visual dynamics but also strengthens cross-modal alignment. This highlights the importance of dense temporal information and long-context processing for achieving robust audiovisual reasoning.

**Open-ended QA vs. MCQ.** To investigate whether the multiple-choice question (MCQ) format overstates model performance, we additionally evaluated several representative models on open-ended question-answering (QA) tasks, where no predefined answer options are provided. In this setting, models must directly generate textual responses, eliminating both the possibility of random guessing and any lexical cues potentially present in candidate options. In Table 4, the accuracy of all models drops significantly compared to their performance on multiple-choice questions. For instance, the Gemini-2.5-Pro, which leads in MCQ benchmarks, experiences a relative accuracy decline exceeding 14 percent in open-ended scenarios, while open-source models exhibit even steeper drops.

Table 4: Comparison of performance on Open-ended Question Answering (QA) and Multiple- Choice Questions (MCQ) across various models.

| Models | Open-ended QA | MCQ |
|---|---|---|
| Gemini-2.0-Flash | 27.06 | 41.50 |
| Qwen2.5-Omni-7B | 17.25 | 29.30 |

## 4 RELATED WORKS

**Omni-Understanding MLLMs.** The development of MLLMs (Chen et al., 2022; Awadalla et al., 2023; Liu et al., 2023; Peng et al., 2025; Yang et al., 2023) began with a foundational focus on

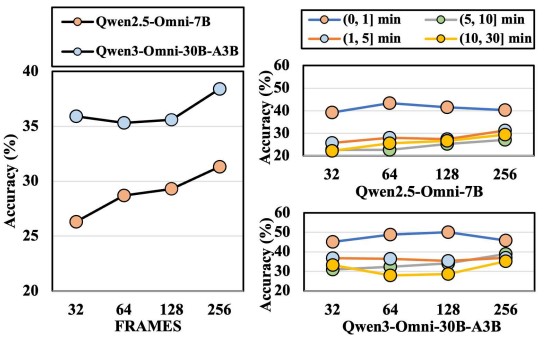 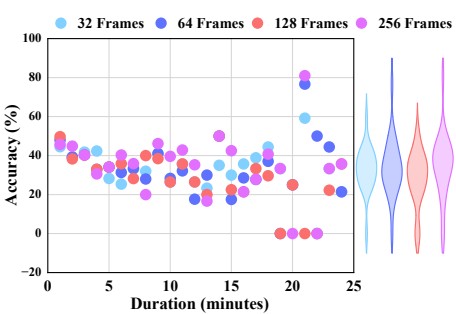

(a) Performance of Qwen2.5-Omni-7B and Qwen3-Omni-30B-A3B at different numbers of frames.

(b) Accuracy of Qwen3-Omni-30B-A3B on questions with videos of varying durations across different numbers of frames.

Figure 7: Performance of selected models when inputting videos with different numbers of frames.

integrating the two primary modalities of vision and language or audio and language. A recent paradigm shift aims to develop Omni-modal MLLMs capable of processing and generating information across an arbitrary combination of modalities ("Any-to-Any"). This approach positions the LLM as a central cognitive engine, unifying diverse data types like image, audio, video, and text within its semantic space (Liu et al., 2024b; Yuan et al., 2025). This has driven a move from integrating pre-trained unimodal components towards developing "natively multimodal" architectures trained from the ground up, as exemplified by models like GPT-4o (Hurst et al., 2024). This ambition is showcased by state-of-the-art models (Xu et al., 2025a; Zhao et al., 2025; Li et al., 2024b; 2025; Yao et al., 2024; Sun et al., 2025; Liu et al., 2025c; Wu et al., 2025a), which pioneer end-to-end streaming capabilities for simultaneously processing video and audio to generate text and speech. At the forefront of this paradigm, proprietary models like Gemini series (Team, 2024; Comanici et al., 2025) demonstrate pinnacle performance, powered by a natively multimodal design and a massive context window that together unlock superior understanding of complex, interwoven data streams.

**MLLM Benchmarks.** The landscape of MLLM evaluation has matured significantly, evolving from foundational perception benchmarks (Liu et al., 2024c; Li et al., 2024a; Yu et al., 2024a;b; Chen et al., 2024a; Jiang et al., 2025) to more sophisticated frameworks (He et al., 2025; Du et al., 2025; Wu et al., 2025b). Recent efforts probe deeper cognitive abilities, with MLLM-Bench (Ge et al., 2025) assessing a hierarchy of cognitive skills. MMMU (Yue et al., 2023) and MMMU-Pro (Yue et al., 2024b) challenging models with expert-level, multi-disciplinary reasoning under stricter protocols like vision-only inputs. Simultaneously, evaluation has specialized into high-stakes domains such as finance (Gan et al., 2024) and medicine (Chen et al., 2024b). For video, some benchmarks (Wang et al., 2019; Li et al., 2021; 2023; Fang et al., 2024; Wu et al., 2024) now focus on the critical challenge of long-context temporal understanding (Liu et al., 2025b), revealing key limitations in current models.

## 5 CONCLUSION

We presented OmniVideoBench, a large-scale benchmark for evaluating audio–visual collaborative reasoning in MLLMs, with diverse videos, carefully verified QA pairs, and explicit reasoning annotations. Experiments show that both open- and closed-source models still struggle with modality complementarity, long-form temporal reasoning, and music understanding, underscoring a large gap from human-level performance. We hope this benchmark will drive future research toward more robust and generalizable multimodal reasoning systems.

## ETHICAL STATEMENTS

This work fully adheres to the ICLR Code of Ethics in all aspects of research conduct. The processes of data collection, usage, annotation, and benchmark construction strictly comply with ethical standards regarding privacy, consent, and responsible AI practices. Videos in OmniVideoBench are strictly limited to academic research purposes. Any form of commercial use is prohibited. All video copyrights remain the property of their original owners. To the best of our knowledge, this study does not involve any data, methodologies, or applications that raise ethical concerns. The authors confirm that they have reviewed and followed the ICLR Code of Ethics throughout the entirety of this research.

## REPRODUCIBILITY STATEMENT

To ensure the reproducibility of our work, we have made the following comprehensive efforts. We provide detailed descriptions of the video collection, filtering, annotation, and reasoning-chain construction pipeline in Sec. 2 and Appendix B. We also present complete statistics of the dataset and explicit definitions of task types, reasoning modalities, and evaluation metrics in Sec. 2.1 and Appendix A. Furthermore, the prompts used in the experiments are documented in detail in Appendix C. Extensive replication experiments demonstrate only minor variations across runs, confirming the stability and full reproducibility of the reported results. We promise to release the experimental code for evaluation in the future to facilitate verification and benchmarking.

## ACKNOWLEDGMENTS

This work was supported by the National Natural Science Foundation of China (No. 62506161). This work was supported in part by the Beijing Major Science and Technology Project (No. Z251100008425023).

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

## A  FULL VIDEO CATEGORY TAXONOMY

Table 5 shows that videos in OmniVideoBench span 8 major categories and 68 subcategories.

Table 5: Full taxonomy of the video dataset.

| Main Category | Subcategories |
|---|---|
| Vlog | Cooking & Cuisine; Travel & Outdoor; Art; Animals; Daily Life at Home; DIY & Handcraft; Gardening; Fitness; Sports; Interviews; Party Games; Makeup & Beauty; Fashion & Styling; Hiking & Trekking |
| News | Politics; Economy; Society; Technology; Education; Healthcare; Military; Law & Justice; Sports; Culture; Entertainment; Weather; Disaster; Transportation |
| Cartoon | 2D Animation; 3D Animation |
| Sports | Basketball; Football (Soccer); Volleyball; Badminton; Table Tennis; Swimming; Figure Skating; Skiing; Gymnastics; Wrestling & Judo; Track & Field; Esports; Others |
| Documentary | Nature & Wildlife; History & Archaeology; Society & Humanity; Politics & Military; Science & Engineering; Medicine & Health; Crime & Law; Art & Culture; Education & Growth; Economy; Environment & Climate; Food & Culinary Culture; Religion & Belief |
| TV | Short; Dramas & Web Series; Variety; Stage Plays; Dance; Mime; Movies |
| Others | Live; Advertisement; Course Replay; Short Video |
| Ego | First-person: People; First-person:Pets |

## B  DETAILED PRINCIPLES OF VIDEO COLLECTION

To ensure an objective and reliable evaluation of MLLMs, the videos included in the benchmark must satisfy multiple requirements, ensuring diversity in both type and duration. The content should provide rich information across audio and visual modalities, while maintaining complementarity between the two. In other words, the benchmark avoids cases where the visual content can be fully inferred from the audio alone, or where the audio is redundant given the visual stream. Furthermore, since many existing video training datasets overlap with the sources of our benchmark—for example, clips from Friends—evaluation may otherwise reduce to simple "answer memorization." To mitigate this unfairness, we additionally consider the publication year of videos when constructing the dataset. The detailed principles for video collection are as follows:

- **Video publication date.** Given that most existing training datasets are constructed from YouTube videos, similar to ours, or contain overlapping content such as identical TV shows, we restrict our selection to videos published after June 2024. We use the most recent videos possible to mitigate unfairness and potential overestimation issues arising from the model having already been exposed to similar content during training.

- **Rich dynamic visual information.** The distinguishing feature of videos compared to images lies in their rich dynamic visual information. A prerequisite for evaluating a model's ability to understand visual information in videos is that the videos themselves contain sufficient dynamic content to be captured and analyzed. Consequently, videos lacking diverse dynamic visual information are excluded, such as those consisting of only several static scenes or perspectives throughout, or those that remain largely static with minimal motion confined to a small corner of the frame.

- **Effective audio information.** In some videos, the audio is completely unrelated to the visual content, such as when only an independent background track is added. We consider

such audio to be invalid. To fairly evaluate the model's capability in audio-visual collaborative reasoning, the audio—whether speech, environmental sound, or music—must align with the visual content.

- **Absence of subtitle.** We excluded videos with embedded subtitles, as such practices convey most of the audio information visually, enabling models to "cheat" through vision alone. Likewise, videos containing large text overlays were regarded as undesirable, since these overlays often directly reveal information about characters' speech, mental states, or ongoing events, thereby undermining the assessment of the model's genuine understanding and reasoning abilities.

- **Video resolution.** To ensure video quality, we require a minimum resolution of 480p, and the visual content must be free from issues such as distortion or blurriness that would hinder comprehension.

## C    PROMPTS USED IN THIS WORK

### C.1    PROMPT FOR OVERALL EVALUATION

> **# Instruction**: You are given a video. Based on the content of the video, answer the following question:
> **# Question**: {Question}
> **# Options**:
> **A**: {Option A} **B**: {Option B} **C**: {Option C} **D**: {Option D}
> **# Task**:
> Answer with the option's letter directly(e.g., A, B, C, or D).
> If your access to the video content is limited, at least one option that is more likely than the others must be chosen.
> Mustn't give any other reason for can not choose!

### C.2    PROMPT TO SELECT QUESTIONS THAT CAN BE ANSWERED WITHOUT RELYING ON OPTIONS

> **# Role**: You are an impartial judge.
> **# Instruction**: Your task is **NOT** to answer the question, but to determine whether the question is inherently **DEPENDENT** on the multiple-choice options in order to be answered.
> **# Task**:
> We aim to convert this multiple-choice question into an open-ended question.
> The video content is **NOT** provided here, but you should assume you have fully watched the video and know everything about it.
> Your job is **ONLY** to decide whether the question itself **\*requires\*** the options to be answerable.
> **# Guidelines**:
>
> - If the question can still be reasonably answered **\*\*without needing the options\*\*** (even if the exact wording might change slightly), return **"No"**.
>
> - If the question cannot be answered at all without the options (e.g., it explicitly asks "Which of the following. . . " ), return **"Yes"**.
>
> **# Question**: {Question}
> **# Answer**: {Answer}
> **Respond ONLY with "Yes" or "No".**

## C.3 PROMPT FOR MULTIPLE-CHOICE QUESTIONS WITH STEP-BY-STEP REASONING

> **# Instruction**: You are given a video. Based on the content of the video, answer the following question:
> **# Question**: {Question}
> **# Options**:
> **A**: {Option A} **B**: {Option B} **C**: {Option C} **D**: {Option D}
> **# Task**:
> Note that you should first reason step by step, and then you should give your final choice in A, B, C, or D.
> Your answer format should be as follows:
> **Step X**: [*Reasoning step X*]
> The final choice is:
> **\bbox**{{*Answer with the option's letter directly(A, B, C, or D).*}}.

## D CASE STUDY

Table 6: Error Distribution Across Models

| Error Category | Gemini2.0-Flash | Qwen3.0-Omni | Qwen2.5-Omni |
|---|---|---|---|
| Audio Understanding | 17.77% | 25.6% | 29.2% |
| Temporal Reasoning | 6.15% | 39.9% | 28.2% |
| Multimodal Fusion | 61.03% | 23.0% | 25.6% |
| Visual Understanding | 12.03% | 8.3% | 9.9% |
| Counting | 1.9% | 2.0% | 4.7% |
| Spatial Understanding | 1.12% | 1.2% | 2.4% |

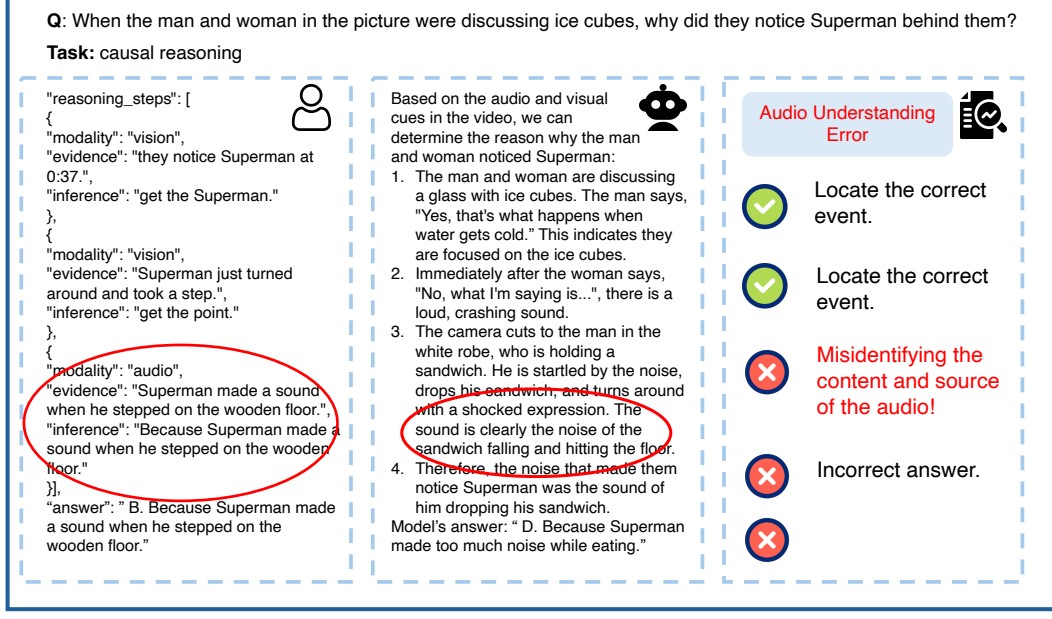

Figure 8: A case about Audio Understanding Error.

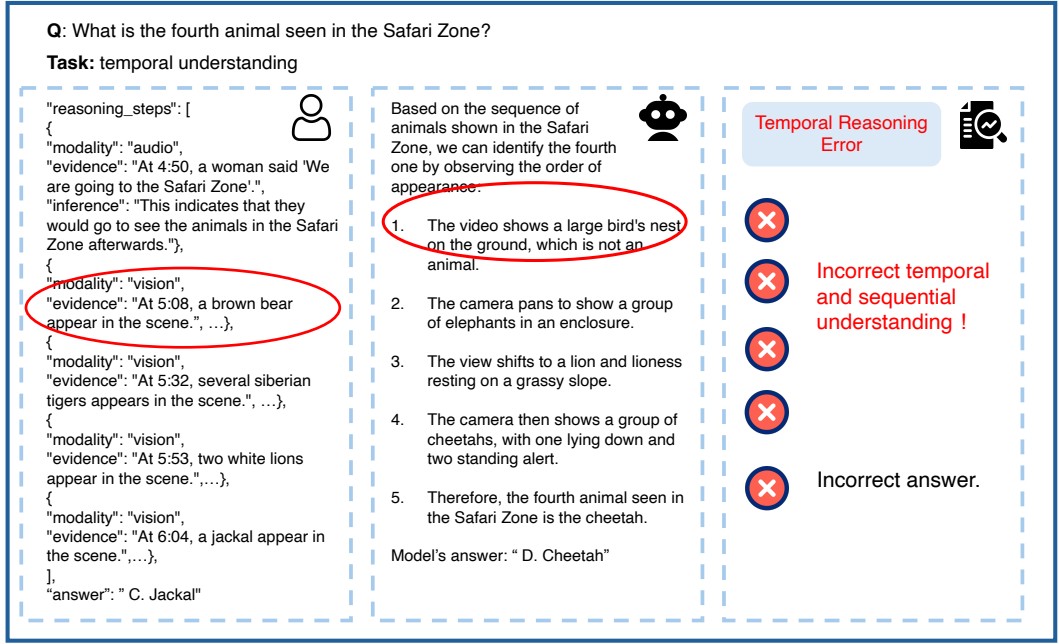

Figure 9: A case about Temporal Reasoning Error.

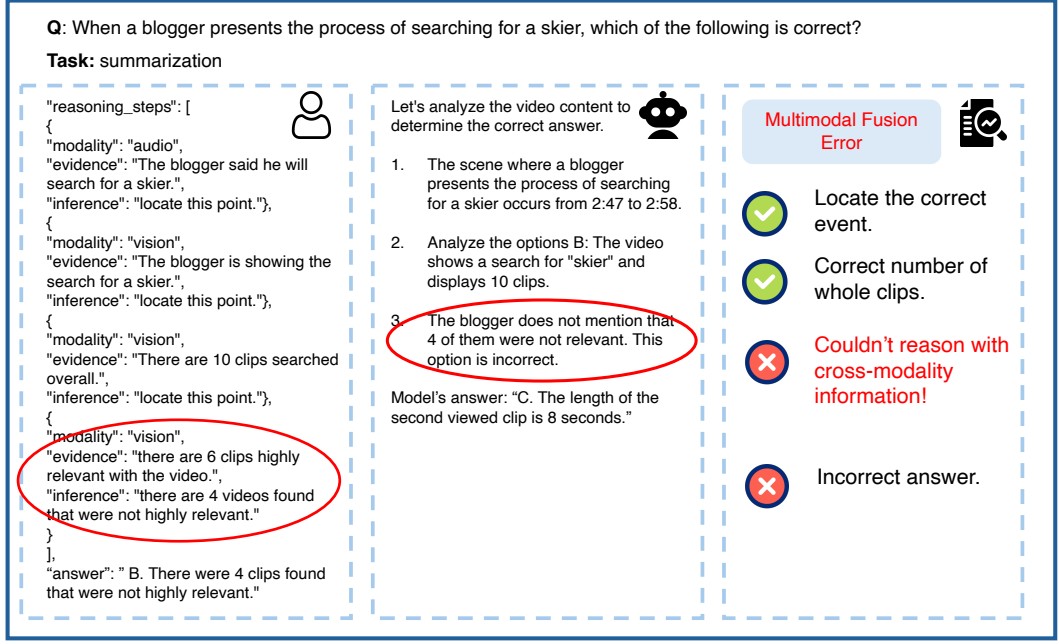

Figure 10: A case about Multimodal Fusion Error.

To better understand the underlying causes of model successes and failures on OmniVideoBench, we conducted a detailed analysis using three representative models: Qwen2.5-Omni, Qwen3-Omni-30A3B, and Gemini2.0-Flash. We aligned each model's incorrect reasoning chain with the OmniVideoBench ground-truth chains using DeepSeek-V3.1 and categorized the resulting discrepancies into 44 concise error types. These types were further grouped into **six high-level categories**: **temporal reasoning error, audio comprehension error, multimodal fusion error, visual comprehension error, counting error, and spatial reasoning error**. A human verification study covering 50 randomly sampled errors per category confirmed a high level of agreement with the automatic

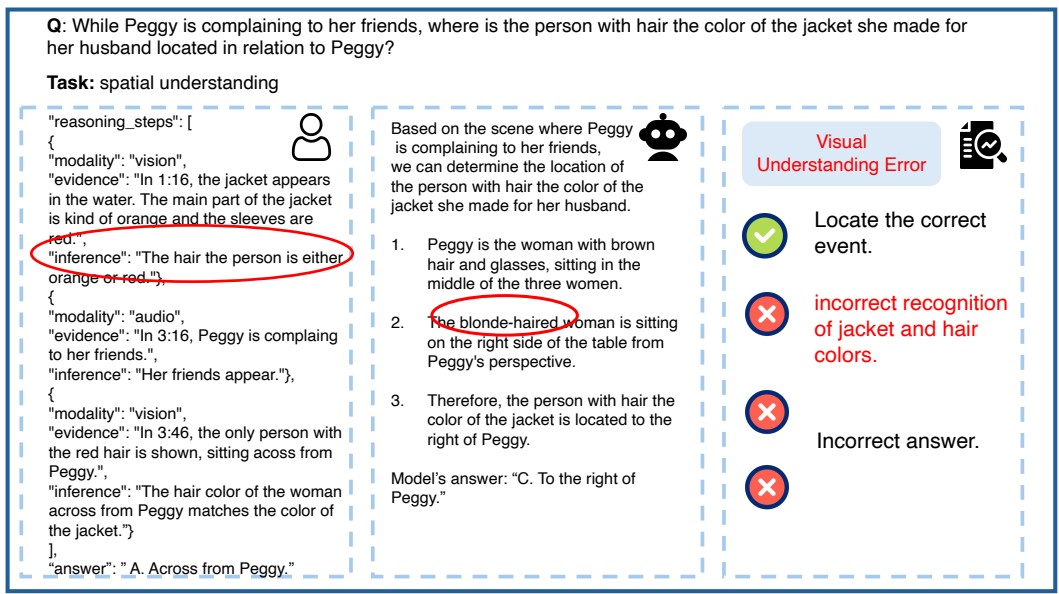

Figure 11: A case about Visual Understanding Error.

annotations. The quantitative comparisons appear in Table 6 and some cases are shown as figs. 8 to 11.

As shown in Figure 12, the first three categories constitute the majority of observed failures for open-source models, indicating that the core bottlenecks of current MLLMs lie in the intersection of *temporal modeling*, *non-speech audio interpretation*, and *cross-modal integration*. These capabilities align closely with the design focus of *OmniVideoBench*, emphasizing long-range temporal dependencies, general acoustic semantics, and robust multimodal alignment.

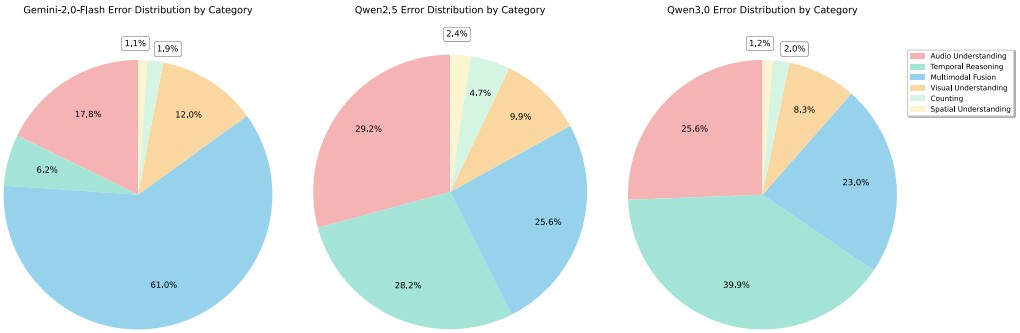

Figure 12: Error distribution of Gemini2.0-Flash, Qwen2.5-Omni and Qwen3-Omni.

**Temporal Reasoning Error.** By analyzing the distribution of error types across different tasks, as shown in Table 7, we find that models frequently struggle with capturing long-range dependencies and sequential relationships, especially when relevant evidence is distributed across multiple clips. Difficulties are further amplified when visual and auditory streams are not perfectly synchronized, requiring the model to integrate misaligned cues over extended durations. These trends suggest that current MLLMs lack sufficiently expressive mechanisms for hierarchical temporal modeling and consistent aggregation of temporal evidence.

**Audio Understanding Error.** By analyzing the distribution of error types across different audio types, as shown in Table 8, we find that non-speech audio—such as music and environmental sounds—remains a major challenge because it lacks stable symbolic anchors comparable to those

Table 7: Error distribution by video duration (percentage) for Gemini2.0-Flash, Qwen2.5-Omni, and Qwen3-Omni.

| Model | Video Duration | Audio Understanding (%) | Temporal Reasoning (%) | Multimodal Fusion (%) | Visual Understanding (%) | Counting (%) | Spatial Understanding (%) |
|---|---|---|---|---|---|---|---|
| Gemini2.0-Flash | (0,1] min | 16.3 | 10.8 | 39.8 | 9.6 | 19.3 | 4.2 |
| | (1,5] min | 14.5 | 10.4 | 39.7 | 9.6 | 18.6 | 7.2 |
| | (5,10] min | 22.5 | 16.7 | 31.3 | 9.3 | 13.2 | 7.0 |
| | (10,30] min | 27.1 | 17.2 | 22.9 | 16.0 | 11.5 | 5.3 |
| Qwen2.5-Omni | (0,1] min | 27.7 | 31.9 | 27.7 | 6.7 | 4.2 | 1.7 |
| | (1,5] min | 23.3 | 36.8 | 24.5 | 10.3 | 3.2 | 2.0 |
| | (5,10] min | 22.9 | 45.3 | 21.2 | 10.0 | 0.0 | 0.6 |
| | (10,30] min | 29.3 | 43.9 | 20.0 | 5.4 | 1.0 | 0.5 |
| Qwen3-Omni | (0,1] min | 25.2 | 31.1 | 30.1 | 7.8 | 5.8 | 0.0 |
| | (1,5] min | 28.0 | 26.8 | 25.9 | 12.1 | 5.9 | 1.3 |
| | (5,10] min | 29.7 | 33.5 | 24.1 | 8.2 | 4.4 | 0.0 |
| | (10,30] min | 33.8 | 25.6 | 25.6 | 10.3 | 3.1 | 1.5 |

in spoken language. Models often fail to map continuous acoustic features (e.g., rhythm, timbre, intensity) to higher-level semantic interpretations involving events, emotions, or actions. Attribution mistakes are also common, including misidentifying background music as speech or incorrectly assigning off-screen sounds to on-screen entities. This reflects the speech-centric nature of existing audio encoders and insufficient grounding in general acoustic semantics.

Table 8: Error distribution by audio type (percentage) for Gemini2.0-Flash, Qwen2.5-Omni, and Qwen3-Omni.

| Model | Audio Type | Audio Understanding (%) | Temporal Reasoning (%) | Multimodal Fusion (%) | Visual Understanding (%) | Counting (%) | Spatial Understanding (%) |
|---|---|---|---|---|---|---|---|
| Gemini2.0-Flash | Music | 50.5 | 9.9 | 14.3 | 2.2 | 17.6 | 5.5 |
| | Sound | 19.0 | 12.9 | 23.1 | 15.6 | 23.1 | 6.1 |
| | Speech | 16.4 | 14.3 | 37.7 | 11.4 | 13.9 | 6.3 |
| Qwen2.5-Omni | Music | 15.9 | 47.8 | 24.6 | 8.7 | 1.4 | 1.4 |
| | Sound | 18.7 | 52.3 | 18.7 | 5.6 | 4.7 | 0.0 |
| | Speech | 28.0 | 36.6 | 23.6 | 8.8 | 1.6 | 1.4 |
| Qwen3-Omni | Music | 23.4 | 28.1 | 31.2 | 9.4 | 7.8 | 0.0 |
| | Sound | 17.8 | 42.1 | 21.5 | 12.1 | 6.5 | 0.0 |
| | Speech | 32.8 | 26.0 | 26.3 | 9.7 | 4.0 | 1.1 |

**Multimodal Fusion Error.** Multimodal fusion failures manifest primarily in two ways: (1) imperfect alignment between visual and auditory cues and (2) *modality neglect*, where the model overrelies on a single modality while disregarding complementary or corrective information present in another. For example, a model may rely solely on visual cues to infer an answer while failing to incorporate crucial auditory evidence. Such issues highlight weaknesses in cross-modal attention robustness and balanced multimodal reasoning.

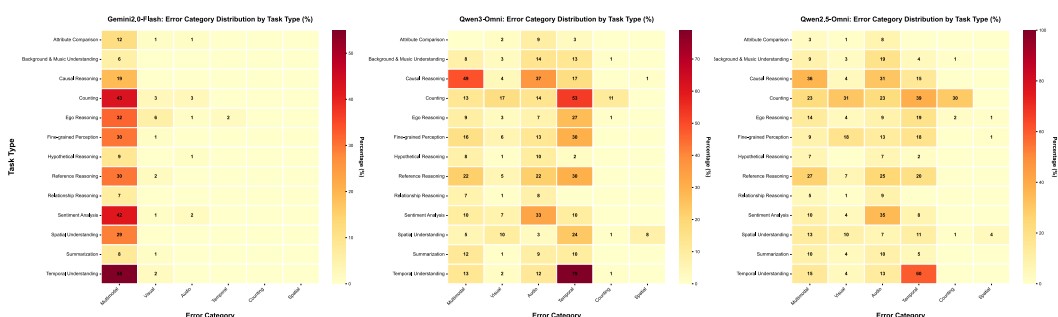

Figure 13: Proportion of error types across different tasks for Gemini2.0-Flash, Qwen2.5-Omni and Qwen3-Omni.

**Counting and Spatial Understanding Error.** Although models demonstrate comparatively weak performance on *Counting* and *Spatial Understanding* tasks, only a limited portion of errors are explicitly attributed to these categories. Further inspection, as shown in Figure 13 reveals that most failures stem from earlier deficits in *fine-grained perception* and *precise temporal localization*. These tasks require detecting multiple small entities, identifying subtle spatial orientations, or localizing objects at specific moments in long video sequences. Errors in early perception or temporal alignment therefore cascade into incorrect counting or spatial reasoning outcomes.

# E    USE OF LLMS

Large language models (LLMs) were utilized in this work solely as research tools to assist with data quality control and ancillary writing support. Specifically, we employed advanced multimodal LLMs (e.g., Gemini-2.0-Flash and DeepSeek-V3) to help filter out questions that could be answered using only a single modality and to identify potential textual biases during dataset refinement, as described in Sec. 2.4. In addition, LLMs were used to perform minor language polishing of the manuscript after the main content was written by the authors. All experimental design, dataset construction, analysis, and conclusions were conceived and executed by the authors without automated decision-making. No confidential, private, or sensitive data were provided to any external LLM services.

