# OpenReview forum: "OmniVideoBench: Towards Audio-Visual Understanding Evaluation for Omni MLLMs"
_ICLR.cc/2026/Conference — ICLR 2026 Poster_

### Official Review · Reviewer_xiBm · 2025-10-30

**Soundness:** 3
**Presentation:** 3
**Contribution:** 3
**Rating:** 6
**Confidence:** 3

**Summary:**

In this work, the authors propose OmniVideoBench, a benchmark for audio–visual understanding in videos. It consists of 1000 QA pairs from 628 videos ranging from several seconds to 30 minutes, within 13 question types, covering temporal reasoning, spatial localization, counting, causal inference, summarization, etc. They also evaluate multiple MLLMs on OmniVideoBench to performance investigation.

**Strengths:**

* Novelty

Currently,  the video understanding community mainly focuses on understanding and reasoning from the visual modality. It ignores the fact that video is the combination of visual and audio clues. This OmniVideoBench provides a bench for comprehensively evaluating reasoning capabilities across both modalities. Hence, it somehow shows the value of this bench for multimodal video understanding.

* Clarity

The paper is well-written with good structure. Hence, the clarity is basically good.

* Significance

This paper focuses on evaluating audio–visual understanding capacity of MLLMs, which is an important and practical problem for video understanding. Hence, the significance is basically OK for video research community.

**Weaknesses:**

* Question Type

The authors choose 13 types for QA pairs. Please further explain why to choose these types. Is it sufficient to evaluate audio–visual understanding in videos?

* Method Insight

It woule be more interesting to investigate or indicate how to design MLLMs to boost the tasks in this benchmark.

* Small Size

The authors collected only 628 original videos. The small number of videos would restrict the generalization of this benchmark.

**Questions:**

Please see the weakness section.

---

> ### Author Response · Authors · 2025-11-20
>
> **Q1: Question Type**
>
> A1: We thank the reviewer for the comment. The selected tasks are designed to reflect the hierarchical structure of **human video understanding**: from l**ow-level perception**, such as fine-grained recognition and attribute comparison, to **mid-level scene understanding**, including spatial reasoning, counting, and temporal comprehension, and further to **high-level cognitive reasoning**, such as causal inference, relational reasoning, summarization, and hypothetical reasoning. This hierarchy aligns with the abilities humans use to process and integrate information.
>
> Additionally, we drew inspiration from classical multimodal video datasets—such as VideoMME, LongVideoBench, and MLVU—and extended their task categorization to audio-visual settings, resulting in tasks like audio-visual ego reasoning. Similarly, we referred to audio-centric datasets, e.g., MMAU, and extended their classifications to form tasks such as audio-visual attribute comparison.
>
> We believe that our 13 task types are sufficient to evaluate audio-visual understanding in videos. Our taxonomy broadly covers the tasks present in existing datasets. To validate this, we applied DeepSeek V3.1 to re-label tasks in WorldSense and AVQA according to our 13 types, allowing the model to assign each instance to one of our categories, with any unclassifiable instances marked as “other.” As shown in the table below, nearly all data can be mapped to our taxonomy, with **virtually absent “other” cases**. This demonstrates that our classification effectively encompasses the range of tasks required for evaluating audio-visual understanding in videos.
>
>
> | Category | WorldSense  | AVQA  | OmniVideoBench(Ours)  |
> |----------|--------------|--------|-------------|
> | Fine-grained Perception | 11.95% | 89.40% | 9.10% |
> | Spatial Reasoning | 8.73% | 1.30% | 6.20% |
> | Attribute Comparison | 2.27% | 0.00% | 2.10% |
> | Background & Music Understanding | 5.96% | 4.50% | 4.70% |
> | Counting | 16.11% | 0.10% | 15.60% |
> | Temporal Understanding | 28.25% | 2.60% | 13.70% |
> | Summarization | 0.95% | 0.80% | 4.90% |
> | Sentiment Analysis | 6.24% | 0.00% | 8.10% |
> | Causal Reasoning | 9.02% | 0.70% | 13.80% |
> | Relationship Reasoning | 5.52% | 0.20% | 2.40% |
> | Reference Reasoning | 3.44% | 0.00% | 9.90% |
> | Ego Reasoning | 0.03% | 0.40% | 7.10% |
> | Hypothetical Reasoning | 1.51% | 0.00% | 2.40% |
> | **Other** | **0.03%** | **0.00%** | **0.00%** |

---

> ### Author Response · Authors · 2025-11-20
>
> **Q2: Method Insight**
>
> A2: Thank you for the constructive suggestion. We agree that understanding how to design MLLMs to better handle the tasks in our benchmark is highly valuable. Although model design is not the primary focus of this work, our analysis suggests several promising directions:
>
> **(1) Large-scale audio–visual co-training with high-quality, modality-synergistic data.**
> Our benchmark reveals that many errors stem from insufficient cross-modal grounding. A key step toward improving MLLMs is constructing *large-scale audio–visual training corpora* where both modalities are strictly required. Automated pipelines with multi-stage filtering—similar to ours—can ensure that audio and visual cues are truly complementary rather than redundant, which encourages models to learn genuine multimodal fusion rather than over-relying on a single modality.
>
> **(2) Improving long-video reasoning through temporal compression and efficient tokenization.**
> Performance degradation on long-duration videos indicates that current models struggle with long-horizon temporal reasoning. Techniques such as **token compression**, **temporal downsampling with semantic preservation**, and **hierarchical memory mechanisms** could reduce redundancy while retaining key events. These approaches have the potential to ease the computational burden and enable MLLMs to maintain coherent reasoning across longer time spans.
>
> **(3) Architectural modifications for tighter audio–visual alignment.**
> Our error analysis highlights failure cases where the model does not properly align auditory and visual streams. Architectural enhancements—e.g., **cross-modal alignment layers**, **shared latent spaces**, or **audio-conditioned visual attention modules**—may help the model integrate cues from both modalities more reliably. Such designs could particularly benefit tasks like audio-visual ego reasoning, attribute comparison, and causal inference.
>
>
> **Q3: Small Size**
>
> A3: Thank you for raising this concern. Although OmniVideoBench contains 628 original videos, this size does not restrict its generalization. The collected videos **cover a broad range of real-world scenarios**, spanning **8** major domains and **68** fine-grained subcategories. This diversity brings variations in visual appearance, audio conditions, temporal structures, and interaction patterns, which we found sufficient to represent typical real-world video distributions.
>
> The final number of videos reflects **deliberate curation** rather than limited collection. We initially gathered **nearly 1,800** raw videos, but many were intentionally removed through a strict multi-stage screening pipeline. Videos with blurred frames, low-density audio information, or overly trivial content were removed early. Additional videos were filtered out during QA-pair annotation, because simple videos usually only yielded trivial questions that did not meet our benchmark requirements. The remaining 628 videos are therefore high-quality, information-dense samples that ensure the reliability of the benchmark.
>
> We additionally conducted **stability experiments** by varying the generation temperature of Qwen2.5-Omni-7B and Qwen3-Omni-30A3B (0.3, 0.5, 0.7, 1.0). Across all settings, the accuracy variance remained **below 0.5**, and the results were stable across different video durations and audio types. This demonstrates that OmniVideoBench provides robust and consistent evaluation, despite its carefully curated size.
>
> ### 📊 By Video Duration
>
> | Duration | Qwen2.5-Omni (Mean) | Qwen2.5-Omni (Variance)| Qwen3-Omni (Mean) | Qwen3-Omni (Variance) |
> |----------|-----------------|---------------------|---------------|-------------------|
> | ≤1 min | 39.33% | 0.09 | 49.56% | 0.09 |
> | 1-5 min | 29.85% | 0.03 | 36.25% | 0.19 |
> | 5-10 min | 21.70% | 0.05 | 38.22% | 0.05 |
> | >10 min | 29.77% | 0.00 | 31.97% | 0.04 |
>
> ### 📊 By Audio Type
>
> | Audio Type | Qwen2.5-Omni (Mean) | Qwen2.5-Omni (Variance)| Qwen3-Omni (Mean) | Qwen3-Omni (Variance) |
> |------------|-----------------|---------------------|---------------|-------------------|
> | Speech | 30.38% | 0.01 | 37.57% | 0.04 |
> | Sound | 29.59% | 0.15 | 41.16% | 0.46 |
> | Music | 23.08% | 0.00 | 34.89% | 0.30 |

---

> ### Author Response · Authors · 2025-11-23
> **Polite follow-up: request for reviewers’ feedback on our responses！**
>
> Hi, we sincerely thank you very much for these **constructive comments and evaluation** of our manuscript. We would like to kindly ask you to **take a look at our responses** and **reevaluate our work** based on our clarifications. Please let us know **whether our response addresses your concerns** or **whether there is any further detail we can provide** to help address these concerns.
>
> Thank you again for dedicating your time to reviewing our paper.

---

> > ### Author Response · Authors · 2025-11-25
> >
> > Dear Reviewer xiBm,
> >
> > I hope this message finds you well. We are writing to gently follow up regarding our rebuttal. We understand that this is a busy period, but if you could spare a moment to review our responses and share any updated evaluation, we would be truly grateful.
> >
> > Please let us know if any additional clarification or information from our side would be helpful. We sincerely appreciate your time and effort in reviewing our work.
> >
> > Warm regards,
> >
> > Authors of OmniVideoBench

---

> > > ### Comment · Reviewer_xiBm · 2025-11-28
> > > **Feedback**
> > >
> > > Thanks for the response. The main concern is addressed. I keep my original rating.

---

### Official Review · Reviewer_iHcc · 2025-10-30

**Soundness:** 2
**Presentation:** 3
**Contribution:** 2
**Rating:** 4
**Confidence:** 3

**Summary:**

This work proposes OmniVideoBench, a new benchmark designed to address key limitations in existing video datasets—namely, the lack of systematic evaluation of audio–visual co-reasoning and inconsistencies in logical task composition. The benchmark is constructed from 628 real-world videos ranging from a few seconds to around 30 minutes, covering three types of audio (speech, sound, and music). It includes 1,000 multiple-choice questions spanning 8 major categories, 68 subcategories, and 13 task types, each accompanied by human-annotated step-by-step reasoning chains.

Experimental results show that current models struggle significantly on this benchmark. While the best model, Gemini-2.5-Pro, only achieves 58.90% accuracy, and most open-source models perform near random. Besides, Long video understanding remains a major challenge for most models. Notably, performance drops sharply for music-dominated audio even for the strongest models.

Overall, OmniVideoBench combines long temporal structure with audio–visual complementarity, providing a more realistic and comprehensive testbed for advancing multimodal video reasoning research.

**Strengths:**

* Broad Coverage and Task Diversity。 The dataset spans 8 high-level categories, 68 subcategories, and 13 task types, with video durations ranging from 4 to 1955 seconds. It also explicitly includes an Ultralong category for videos longer than 10 minutes.
* Step-by-Step Human-Annotated Reasoning Chains. Each question is accompanied by a human-labeled step-by-step reasoning chain, in terms of modality, evidence, and inference triples.
* Data Quality Control. The authors take multiple steps to ensure data quality, including a three-stage filtering pipeline and the exclusion of videos with large-scale on-screen subtitles that might leak answers or bias model predictions.
* Comprehensive Evaluation. The benchmark is evaluated using a wide range of models, including both open-source and close-source of varying scales.

**Weaknesses:**

* Limited Per-Task Coverage. The dataset contains 1,000 questions spread across 13 distinct task types, resulting in fewer than 100 examples per task on average. This limited coverage may constrain the robustness of task-specific evaluation and generalization analysis.
* Unbalanced Audio Category Distribution. The distribution of audio types is heavily skewed—Speech accounts for over three-quarters of the dataset, while Music constitutes only 9.1%. This imbalance may bias models and limit insights into performance under underrepresented audio conditions.
* Lack of Human Baseline. Although the conclusion emphasizes a large gap between human and model performance, no human baseline is reported in the experiments.
* Dataset unavailable. The authors did not provide the dataset link in the paper.

**Questions:**

1. What is the core distinction between this work and WorldSense that is mentioned in Table 1?
Table 1 suggests that the two benchmarks share similar characteristics across several dimensions—including modality coverage, domain diversity, video type, audio type, and answer format. A more explicit comparison would help clarify the novel contributions of OmniVideoBench.
2. Qwen2.5-Omni performs better on WorldSense than on OmniVideoBench. Does this imply that OmniVideoBench presents a more difficult challenge?
Beyond video length, are there other factors or task-level distinctions that contribute to the increased difficulty of OmniVideoBench? Providing such an analysis would help justify the benchmark’s added value.
3. What is the significance of emphasizing long videos?
Table 1 shows that SOTA model performance is comparable between the 5–10 minute and 10+ minute video subsets. Additionally, WorldSense already includes videos of up to 10 minutes. Could the authors clarify what unique challenges OmniVideoBench introduces with longer videos?

---

> ### Author Response · Authors · 2025-11-20
>
> **Q1: Limited Per-Task Coverage.**
>
> A1: We appreciate this thoughtful question. While OmniVideoBench indeed contains 13 carefully designed task types, we argue that the per-task analyses remain statistically meaningful and diagnostically valid, for (1) **Our task types comprehensively cover essential audio-visual understanding challenges**. The 13 types were derived from a systematic analysis of multimodal reasoning demands in authentic video scenarios, targeting fundamental cognitive capabilities rather than arbitrary categories. Each type represents a distinct, necessary dimension for genuine audio-visual comprehension. (2) **Consistent Cross-Model Trends Indicate Stability**. As shown in Figure 5 in our paper, closed-source and open-source models exhibit consistent relative rankings across all 13 tasks (e.g., Gemini-2.5-Pro > Gemini-2.0-Flash > Qwen3-Omni-30A3B). These stable performance hierarchies demonstrate that observed differences reflect inherent task difficulty rather than sampling artifacts. (3) **Task Types Are Analytical Dimensions, Not Training Classes**. OmniVideoBench serves diagnostic evaluation, not statistical estimation. Each type isolates a specific reasoning dimension—temporal, causal, spatial, or emotional—where high-quality, manually verified instances (average 5.68 reasoning steps per question) effectively reveal model limitations. Small but semantically controlled subsets suffice for comparative analysis, as validated in established benchmarks like MMBench and MMMU-Pro.
>
> Additionally, our ablation experiments modified the generation temperature of Qwen2.5-Omni-7B and Qwen3-Omni-30A3B to 0.3, 0.5, 0.7, and 1.0, respectively. We observed that the accuracy variance across all task categories **did not exceed 1.0**, demonstrating that the OmniVideoBench task configuration provides a **stable and robust** evaluation solution for model capability assessment.
>
> ### 📊 By Question Type
>
> | Question Type | Qwen2.5-Omni-7B-Mean | Qwen2.5-Omni-7B-Variance | Qwen3-Omni-30A3B-Mean | Qwen3-Omni-30A3B-Variance |
> |---------------|-----------------|---------------------|---------------|-------------------|
> | Reference Reasoning | 26.26% | 0.00 | 33.33% | 0.00 |
> | Spatial Understanding | 19.35% | 0.00 | 24.19% | 0.00 |
> | Counting | 20.83% | 0.14 | 27.40% | 0.10 |
> | Relationship Reasoning | 45.83% | 0.00 | 54.17% | 0.00 |
> | Temporal Understanding | 28.83% | 0.18 | 33.21% | 0.53 |
> | Sentiment Analysis | 35.80% | 0.00 | 50.62% | 0.00 |
> | Hypothetical Reasoning | 16.67% | 0.00 | 41.67% | 0.00 |
> | Causal Reasoning | 34.06% | 0.00 | 40.40% | 0.13 |
> | Fine-grained Perception | 29.67% | 0.00 | 37.36% | 0.00 |
> | Attribute Comparison | 33.33% | 0.00 | 41.67% | 5.67 |
> | Ego Reasoning | 40.85% | 0.00 | 56.34% | 0.00 |
> | Summarization | 42.86% | 0.00 | 52.55% | 1.04 |
> | Background & Music Understanding | 23.40% | 0.00 | 29.79% | 0.00 |
>
> ### 📊 By Video Duration
>
> | Duration | Qwen2.5-Omni-7B-Mean | Qwen2.5-Omni-7B-Variance | Qwen3-Omni-30A3B-Mean | Qwen3-Omni-30A3B-Variance |
> |----------|-----------------|---------------------|---------------|-------------------|
> | ≤1 min | 39.33% | 0.09 | 49.56% | 0.09 |
> | 1-5 min | 29.85% | 0.03 | 36.25% | 0.19 |
> | 5-10 min | 21.70% | 0.05 | 38.22% | 0.05 |
> | >10 min | 29.77% | 0.00 | 31.97% | 0.04 |
>
> ### 📊 By Audio Type
>
> | Audio Type | Qwen2.5-OmniMean | Qwen2.5-OmniVariance | Qwen3-OmniMean | Qwen3-OmniVariance |
> |------------|-----------------|---------------------|---------------|-------------------|
> | Speech | 30.38% | 0.01 | 37.57% | 0.04 |
> | Sound | 29.59% | 0.15 | 41.16% | 0.46 |
> | Music | 23.08% | 0.00 | 34.89% | 0.30 |

---

> ### Author Response · Authors · 2025-11-20
>
> **Q2: Unbalanced Audio Category Distribution.**
>
> A2: We appreciate your observation regarding the distribution of audio types. It is important to clarify at the outset that the labeling in OmniVideoBench is based on the type of audio information required to answer a question, rather than the full composition of the video’s audio track. For example, in a musical theatre video, if the question concerns a spoken announcement by the host, the item is labeled as Speech rather than Music. This naturally results in fewer Sound and Music labels, even though these audio elements may be present in the underlying video.
>
> Also, the distribution OmniVideoBench obtains is also consistent with real-world audio usage patterns. Using Qwen3-Omni-30A3B, we re-annotated tasks in WorldSense according to the audio modality needed to answer each question. The resulting ratio is roughly **70.99 : 17.97 : 11.04** for Speech, Sound, and Music, while the corresponding distribution in OmniBench is **67.51 : 23.21 : 9.28**. These independent datasets suggest that such proportions are characteristic of open-domain video content.
> | Benchmark      | Speech            | Sound       | Music       |
> | :-------------- | :----------  | :---------- | :----------- |
> | **WorldSense** | 70.99%  | 17.97% | 11.04% |
> | **OmniBench** | 67.51%| 23.21% |  9.28% |
> | **OmniVideoBench(Ours)** | 76.20%  | 14.70% | 9.10% |
>
> It is also important to note that the current distribution appears only after our multi-stage filtering pipeline. The initial pool of approximately 2,500 annotated questions exhibited a considerably more balanced distribution, with proportions of **45.04 : 28.68 : 26.28** for Speech, Sound, and Music, respectively. Many Sound and Music questions were removed because they lacked meaningful audio–visual interaction or could be answered trivially without multimodal reasoning. The retained questions therefore reflect high-quality, genuinely multimodal items rather than an intrinsic bias in the dataset.
>
> Moreover, the present proportions do not compromise evaluation **stability**. We further conducted a stability experiment by modifying the generation temperature of Qwen2.5-Omni-7B and Qwen3-Omni-30A3B to 0.3, 0.5, 0.7, and 1.0, respectively. The accuracy statistics across the three audio categories are shown in the table below. We observed that the variance in accuracy across all audio types remained within **0.5**, indicating that the audio configuration of OmniVideoBench provides a stable and robust evaluation protocol for assessing model capabilities.
>
> We will include these clarifications and the full quantitative details in the revised manuscript.
>
> | Audio Type | Qwen2.5-Omni (Mean) | Qwen2.5-Omni (Variance)| Qwen3-Omni (Mean) | Qwen3-Omni (Variance) |
> |------------|-----------------|---------------------|---------------|-------------------|
> | Speech | 30.38% | 0.01 | 37.57% | 0.04 |
> | Sound | 29.59% | 0.15 | 41.16% | 0.46 |
> | Music | 23.08% | 0.00 | 34.89% | 0.30 |

---

> ### Author Response · Authors · 2025-11-20
>
> **Q3: Lack of Human Baseline.**
>
> A3: Thank you for highlighting the need for a human baseline. In response, we conducted a dedicated human evaluation designed to be **stable**, **unbiased**, and **aligned with the benchmark setting**.
>
> We invited 10 qualified annotators, including 8 **graduate students** experienced in multimodal research and 2 experts trained in **music-related analysis**. Prior to formal evaluation, we demonstrated through random sampling tests that human assessment exhibits only **minimal bias** and is feasible. We consolidated all questions requiring **Musical Knowledge** and assigned them to two music experts. The remaining questions were distributed roughly equally between **Speech** and **Sound** samples, then evenly assigned to highly qualified personnel such as graduate students to achieve persuasive results. Human responses were conducted without time constraints, achieving a final accuracy rate of **82.7%** and the results for audio classification are shown in the table below. This indicates that existing models still exhibit a **significant gap** compared to human capabilities.
>
> |Model        | Speech(%) | Sound(%) | Music(%) | <1 min(%) | 1-5 mins(%) | 5-10 mins(%) | >10 mins(%) | Avg.(%) |
> |------------------|------------------|------------------|--------------|---------------|--------------|--------------|--------------|--------------|
> | Qwen2.5-Omni-7B      | 30.70 | 25.33 | 23.07 | 41.57 | 27.41 | 25.33 | 26.72 | 29.30 |
> |**Human Baseline** | 84.12 | 74.73 | 80.27 | 78.31 | 85.71 | 80.79 | 83.21 | 82.70 |
>
> Thank you for your suggestion. We have incorporated the human benchmark results into the revised version.

---

> ### Author Response · Authors · 2025-11-20
>
> **Q4: Dataset unavailable.**
>
> A4: Thank you for bringing this issue to our attention. Given that OmniVideoBench's videos are sourced from platforms like YouTube and processed through editing, we are currently unable to publicly release our dataset and videos in the paper due to **copyright restrictions**. However, you may find several **sample videos and questions** in our anonymized repository: https://anonymous.4open.science/r/OmniVideoBench-F44D
>
> For anonymization purposes, all author-identifying information has been removed from the repository.

---

> > ### Comment · Reviewer_iHcc · 2025-11-26
> > **Response to authors**
> >
> > I appreciate the effort of the authors. The authors addressed most of my concerns. But the lack of dataset availability is unacceptable for a benchmark paper. Do the authors have any plan to improve the transparency and dataset availability?
> >
> > Sincerely,

---

> > > ### Author Response · Authors · 2025-11-26
> > >
> > > We sincerely thank the reviewer for their appreciation and constructive feedback. Regarding dataset availability, we must clarify that due to copyright restrictions from sourcing videos from platforms like YouTube and subsequent processing, our dataset is similarly accessible only upon request via a questionnaire and email approval, aligning with the access protocols of established benchmarks like VideoMME [1], Ego4D [2], and LSMDC [3]. To enhance transparency, we have provided five sample videos and the complete QA list in the `example` folder of our anonymized repository (https://anonymous.4open.science/r/OmniVideoBench-F44D), along with all evaluation code; we are committed to continuously evaluating the latest models and maintaining a public leaderboard.
> > >
> > > [1] Fu, C., Dai, Y., Luo, Y., Li, L., Ren, S., Zhang, R., ... & Sun, X. (2025). Video-mme: The first-ever comprehensive evaluation benchmark of multi-modal llms in video analysis. In Proceedings of the Computer Vision and Pattern Recognition Conference (pp. 24108-24118).
> > >
> > > [2] Grauman, K., Westbury, A., Byrne, E., Chavis, Z., Furnari, A., Girdhar, R., ... & Malik, J. (2022). Ego4d: Around the world in 3,000 hours of egocentric video. In Proceedings of the IEEE/CVF conference on computer vision and pattern recognition (pp. 18995-19012).
> > >
> > > [3] Rohrbach, A., Torabi, A., Rohrbach, M., Tandon, N., Pal, C., Larochelle, H., ... & Schiele, B. (2017). Movie description. International Journal of Computer Vision, 123(1), 94-120.

---

> > > > ### Comment · Reviewer_iHcc · 2025-11-26
> > > >
> > > > Thanks for the clarification. I have adjusted the score accordingly.

---

> ### Author Response · Authors · 2025-11-20
>
> **Q5:  The distinction between OmniVideoBench and WorldSense.**
>
> A5: Thank you for posing this insightful question. While OmniVideoBench and WorldSense share several similarities in terms of modality coverage and domain diversity, **their core objectives** and **design philosophies differ significantly**, as outlined in the table below. While WorldSense also includes video clips exceeding 10 minutes, it primarily targets perception tasks on **medium-to-short videos**, with an average duration of just 141.1 seconds. In contrast, OmniVideoBench features longer videos (average 384.24 seconds) and requires models to **integrate visual and auditory clues in a logically consistent manner**, ensuring each question requires an average of **5.68 reasoning steps** to answer. This explicit cross-modal reasoning supervision mechanism enables OmniVideoBench to not only evaluate answer correctness but also test whether models possess **logically coherent reasoning processes**. Additionally, we employ a rigorous question design process to ensure each question can only be answered by **integrating audio and video modalities**. For instance, when audio information is removed, Qwen2.5-Omni-7B's accuracy **drops to 26.4%**, approaching **random performance (25%)**. This demonstrates the **challenging nature** of OmniVideoBench in terms of **modal integration and reasoning**.
>
> | Benchmark        | Avg. Video Duration (s) | Video Subcategories | Avg. Reasoning Steps | Qwen2.5-Omni-7B full (%) | Qwen2.5-Omni-7B w/o audio (%) |
> |------------------|--------------------------|----------------------|------------------------|----------------------------|--------------------------------|
> | WorldSense       | 141.1                   | 67                   | /                      | 45.2                       | 39.2                           |
> | OmniVideoBench   | 384.24                  | 68                   | 5.68                   | 29.3                       | 26.4                           |

---

> ### Author Response · Authors · 2025-11-20
>
> **Q6: Factors Contributing to the Higher Difficulty of OmniVideoBench**
>
> A6: We appreciate the reviewers' insights. Qwen2.5-Omni does indeed outperform OmniVideoBench in the WorldSense test, but this difference reflects the **inherently greater difficulty of OmniVideoBench**. The test design significantly increases the challenge across multiple dimensions, with complexity far exceeding mere video duration. First, the videos in OmniVideoBench feature **substantially longer durations and more intricate temporal structures**: many tasks require tracking events across vast time intervals, establishing cross-scene causal relationships, or extracting overarching narrative hierarchies.
>
> Furthermore, as noted in Problem 1, each OmniVideoBench question requires an **average of 5.68 reasoning steps** to arrive at the correct answer. This means that for models, possessing strong video comprehension alone is insufficient; they must also maintain **logical consistency and coherence** throughout the video understanding process—an aspect currently overlooked by many MLLMs.
>
> More crucially, OmniVideoBench is specifically designed for **audiovisual collaborative reasoning**. Our ablation experiments show that when audio input is removed, Qwen2.5-Omni-7B's accuracy **drops from 29.3% to 26.4%, approaching random performance**, whereas in WorldSense the model still correctly answered 39.2% of questions without audio. This demonstrates that OmniVideoBench requires models to **rely on both modalities simultaneously**, posing a deeper challenge to a model's ability to integrate multimodal information.
>
> | Benchmark        | Qwen2.5-Omni-7B full (%) | Qwen2.5-Omni-7B w/o audio (%) | Avg. Reasoning Steps |
> |------------------|---------------------------|--------------------------------|------------------------|
> | WorldSense       | 45.2                      | 39.2                           | /                      |
> | **OmniVideoBench** | **29.3**                  | **26.4**                       | **5.68**               |

---

> ### Author Response · Authors · 2025-11-20
>
> **Q7: The Significance of Long Videos in OmniVideoBench.**
>
> A7: Thank you for raising this important question. OmniVideoBench's emphasis on long videos extends beyond merely increasing duration; it introduces **qualitative challenges** that current benchmarks like WorldSense do not fully address. Long videos inherently contain **richer, more dispersed, and less coherent information**. Many problems in OmniVideoBench rely on understanding events separated by large time intervals, integrating **causal clues across scenes**, or recognizing **global narrative structures**. This makes **long-range dependency modeling** a core requirement, rather than a byproduct of video length. In practice, such long-term sequence reasoning frequently appears in real-world scenarios (e.g., vlogs, documentaries, multi-character interactions) but remains a major challenge for current LLMs and MLLMs.
>
> Regarding the reviewer’s observation that state-of-the-art models show similar accuracy on the 5–10 minute subset and the >10 minute subset, this reflects **model-specific characteristics rather than insufficient difficulty**. For instance, Gemini-2.5-Pro uses fixed 1 fps sampling and possesses strong sequence modeling ability, enabling it to maintain global coverage even for ultra-long videos. However, when long-sequence capabilities weaken, **the duration gradient becomes visible**: Gemini-2.0-Flash drops **from 41.05% (5–10 min) to 34.87% (>10 min)**. Open-source models such as Qwen2.5-Omni-7B fail **once video length exceeds 1 minute**, remaining uniformly low across all medium-to-long subsets. Thus, OmniVideoBench’s **duration-structured design differentiates strong and weak models**: strong models show a smooth decay curve, while weaker models collapse early.
> Compared to WorldSense, OmniVideoBench presents **unique long-form challenges** by requiring **joint use of audio and visual information across extended time spans**. Every question is constructed such that **both modalities are necessary** and cues must be integrated over dispersed temporal segments. OmniVideoBench also excludes questions solvable with a single modality or short-range cues, ensuring that longer temporal spans translate into **substantive reasoning difficulty**, not simply longer inputs.
>
> Therefore, the contribution of emphasizing long videos lies not in duration itself but in how OmniVideoBench **transforms long videos into structurally demanding audio–visual reasoning problems**. This design exposes model limitations, especially in **long-term temporal integration** and **cross-modal consistency, and provides **fine-grained diagnostic signals** for advancing future MLLMs.

---

> ### Author Response · Authors · 2025-11-23
> **Polite follow-up: request for reviewers’ feedback on our responses!**
>
> Hi, we sincerely thank you very much for these **constructive comments and evaluation** of our manuscript. We would like to kindly ask you to **take a look at our responses** and **reevaluate our work** based on our clarifications. Please let us know **whether our response addresses your concerns** or **whether there is any further detail we can provide** to help address these concerns.
>
> Thank you again for dedicating your time to reviewing our paper.

---

> > ### Author Response · Authors · 2025-11-25
> >
> > Dear Reviewer iHcc,
> >
> > I hope this message finds you well. We are writing to gently follow up regarding our rebuttal. We understand that this is a busy period, but if you could spare a moment to review our responses and share any updated evaluation, we would be truly grateful.
> >
> > Please let us know if any additional clarification or information from our side would be helpful. We sincerely appreciate your time and effort in reviewing our work.
> >
> > Warm regards,
> >
> > Authors of OmniVideoBench

---

### Official Review · Reviewer_rRon · 2025-10-31

**Soundness:** 3
**Presentation:** 2
**Contribution:** 2
**Rating:** 4
**Confidence:** 4

**Summary:**

This paper introduces OmniVideoBench, a large-scale benchmark designed to evaluate the collaborative audiovisual reasoning capabilities of multimodal large language models. The benchmark comprises 1,000 manually annotated high-quality question-answer pairs (QA) across 628 videos, each featuring explicit step-by-step reasoning chains indicating modalities and evidence. OmniVideoBench spans 8 primary video genres, 68 subcategories, and 13 distinct task types (e.g., temporal, spatial, causal reasoning), structured to comprehensively evaluate modal complementarity and logical consistency. The paper evaluates both open-source and proprietary MLLMs on OmniVideoBench, revealing that model performance lags significantly behind human capabilities, particularly on tasks requiring genuine multimodal integration.

**Strengths:**

1. The paper ensures the richness and coverage of the dataset, comprising 1,000 distinct QA pairs that span a wide range of real-world scenarios, video durations, and audio types. It also contains 13 different task types covering diverse reasoning skills.
2. The annotation protocol ensures that all questions require true audio-visual integration and stepwise reasoning, with multi-layered filtering to weed out unimodal or bias-prone items.

**Weaknesses:**

1. The paper does not provide statistical results comparing it with other relevant datasets, which fails to highlight the contribution of the proposed dataset.
2 From Table 1 and Figure 3, it is evident that the vast majority of QA pairs relate to Speech (76.2%) versus Sound (14.7%) and Music (9.1%), creating considerable class imbalance.
3 The benchmark's positioning relative to several directly analogous or recently proposed audio-visual (AV) benchmarks is incomplete. Recent works, such as AVHBench (Kim et al., 2025) and DAVE (Radevski et al., 2024), are not cited or discussed, nor are specialized audio-visual QA datasets, including AVQA (Yang et al., 2022) and MusicAVQA (Li et al., 2022).
4. While human annotation is used in construction, the paper does not report human baseline accuracy or response variability for the main test set.
5. While Figure 1 gives some specific sample breakdowns, the paper lacks a deeper set of qualitative analyses of successful versus failure cases, especially for (a) long video cases and (b) music understanding tasks.

**Questions:**

1. Can the authors provide quantitative statistics on the efficacy of the automated filtering steps in Section 2.4? Specifically, what is the rejection or retention rate at each filtering stage, and what percentage of QA pairs end up truly requiring both modalities?
2. Please clarify how "semantic units" ($S_i$) are operationalized for the semantic distance metric. Is this manual phrase decomposition, or is some NLP toolchain applied? This is crucial to evaluating distractor design reproducibility.
3. Will human benchmark results (e.g., accuracy, agreement rates) on OmniVideoBench be reported?

---

> ### Author Response · Authors · 2025-11-20
>
> Thank you for your suggestions!
>
> **W1.1: Statistic comparison results**
>
> A1.1: We have added more detailed quantitative comparisons and updated the comparison results to Table 2 of the paper. The updated table is as follows. Current MLLMs have demonstrated strong capabilities in short-video understanding and question-answering, with models like Qwen2.5Omni-7B achieving a score of **74.7** on **AVHBench (Judgment)**. However, there remains a lack of discriminative evaluation methods for **long-video** understanding. **OmniVideoBench** fills this gap by comprehensively evaluating MLLMs' **cross-temporal understanding** and **audiovisual collaborative reasoning** capabilities through video question-answering tasks spanning up to **1955 seconds**, with an average duration of **384.24 seconds**. Additionally, unlike benchmarks focused on **specific problem** types—such as **DAVE**[4], which concentrates on the collaborative understanding of visual and environmental sound, OmniVideoBench prioritizes evaluating models' **comprehensive capabilities** in audiovisual collaborative reasoning. Consequently, our dataset incorporates over **68 video subcategories** and **13 task types**, with audio spanning Speech, Sound, and Music. We ensure each problem requires integrating information from **two modalities** to solve, enabling a holistic assessment of model capabilities.
> You can find more information in section 2.6 DATASET COMPARISON in the **last version of our paper**.
>
> | Benchmark              | Modality | Qwen2.5-Omni | Multiple Domains | Video Type | Audio Type   | Video Duration | Answer Type |
> |------------------------|----------|--------------|------------------|------------|--------------|----------------|-------------|
> | AVQA                   | V+A      | /            | ✗                | R          | So           | 10             | MC          |
> | Music-AVQA            | V+A      | /            | ✗                | R+S        | Mu           | 60             | CLS         |
> | AVTRUSTBENCH          | V+A      | /            | ✓                | R+S        | Sp+So+Mu     | 10/60          | MC          |
> | MMAU                   | A        | 71.0         | ✓                | /          | Sp+So+Mu     | /              | MC          |
> | DAVE                   | V+A      | 31.0         | ✓                | R+S        | So           | ≤60            | MC          |
> | AV-Odyssey             | I+A      | /            | ✓                | R          | Sp+So+Mu     | /              | MC          |
> | AVHBench (Judgement)   | V+A      | 74.7         | ✓                | R+S        | So           | 10             | CLS         |
> | OmniBench              | I+A      | 56.1         | ✓                | R          | Sp+So+Mu     | /              | MC          |
> | Daily-Omni             | V+A      | 47.5         | ✗                | R          | Sp+So+Mu     | 30/60          | MC          |
> | WorldSense             | V+A      | 48.3         | ✓                | R          | Sp+So+Mu     | 15–656         | MC          |
> | **OmniVideoBench (Ours)** | **V+A**  | **29.3**     | **✓**            | **R**      | **Sp+So+Mu** | **4–1955**     | **MC**       |

---

> ### Author Response · Authors · 2025-11-20
>
> **W1.2: Imbalanced audio type distribution**
>
> A1.2: We appreciate your observation regarding the distribution of audio types. It is important to clarify at the outset that the labeling in OmniVideoBench is based on the type of audio information required to answer a question, rather than the full composition of the video’s audio track. For example, in a musical theatre video, if the question concerns a spoken announcement by the host, the item is labeled as Speech rather than Music. This naturally results in fewer Sound and Music labels, even though these audio elements may be present in the underlying video.
>
> Also, the distribution OmniVideoBench obtains is also consistent with real-world audio usage patterns. Using Qwen3-Omni-30A3B, we re-annotated tasks in WorldSense according to the audio modality needed to answer each question. The resulting ratio is roughly **70.99 : 17.97 : 11.04** for Speech, Sound, and Music, while the corresponding distribution in OmniBench is **67.51 : 23.21 : 9.28**. These independent datasets suggest that such proportions are characteristic of open-domain video content.
> | Benchmark      | Speech            | Sound       | Music       |
> | :-------------- | :----------  | :---------- | :----------- |
> | **WorldSense** | 70.99%  | 17.97% | 11.04% |
> | **OmniBench** | 67.51%| 23.21% |  9.28% |
> | **OmniVideoBench(Ours)** | 76.20%  | 14.70% | 9.10% |
>
> It is also important to note that the current distribution appears only after our multi-stage filtering pipeline. The initial pool of approximately 2,500 annotated questions exhibited a considerably more balanced distribution, with proportions of **45.04 : 28.68 : 26.28** for Speech, Sound, and Music, respectively. Many Sound and Music questions were removed because they lacked meaningful audio–visual interaction or could be answered trivially without multimodal reasoning. The retained questions therefore reflect high-quality, genuinely multimodal items rather than an intrinsic bias in the dataset.
>
> Moreover, the present proportions do not compromise evaluation **stability**. We further conducted a stability experiment by modifying the generation temperature of Qwen2.5-Omni-7B and Qwen3-Omni-30A3B to 0.3, 0.5, 0.7, and 1.0, respectively. The accuracy statistics across the three audio categories are shown in the table below. We observed that the variance in accuracy across all audio types remained within **0.5**, indicating that the audio configuration of OmniVideoBench provides a stable and robust evaluation protocol for assessing model capabilities.
>
> We will include these clarifications and the full quantitative details in the revised manuscript.
>
> | Audio Type | Qwen2.5-Omni (Mean) | Qwen2.5-Omni (Variance)| Qwen3-Omni (Mean) | Qwen3-Omni (Variance) |
> |------------|-----------------|---------------------|---------------|-------------------|
> | Speech | 30.38% | 0.01 | 37.57% | 0.04 |
> | Sound | 29.59% | 0.15 | 41.16% | 0.46 |
> | Music | 23.08% | 0.00 | 34.89% | 0.30 |

---

> ### Author Response · Authors · 2025-11-20
>
> **W1.3: Comparison with relavent datasets**
>
> A: Thank you for your insightful suggestions. We recognize that the benchmarks you mentioned are **highly relevant** to our work and have added related discussions **in the latest version**.As classic datasets in the audiovisual domain, **AVQA**[1] and **Music-AVQA**[2] have profoundly influenced our work, particularly in video selection and question type design. We have acknowledged their contributions in our latest paper. Emerging audiovisual benchmarks like **AVHBench**[3] primarily focus on evaluating fine-grained understanding capabilities for **short videos (under 60 seconds)**, providing valuable guidance for multimodal large language models in the domains of fine-grained detection and audiovisual collaborative reasoning. **DAVE**[4] and other benchmarks focus on evaluating **specific aspects** of MLLMs' capabilities (such as diagnostic audio-visual evaluation). In contrast, **OmniVideoBench** innovatively introduces the challenge of **collaborative audio-visual reasoning for long videos**, providing a **comprehensive and discriminative evaluation** of multimodal models' video understanding capabilities.
>
> In the latest revision of our paper, we have incorporated citations to the relevant benchmarks and expanded the corresponding discussion accordingly.
>
> [1] Yang P, Wang X, Duan X, et al. Avqa: A dataset for audio-visual question answering on videos[C]//Proceedings of the 30th ACM international conference on multimedia. 2022: 3480-3491.
>
> [2] Li G, Wei Y, Tian Y, et al. Learning to answer questions in dynamic audio-visual scenarios[C]//Proceedings of the IEEE/CVF conference on computer vision and pattern recognition. 2022: 19108-19118.
>
> [3] Sung-Bin, Kim, et al. "Avhbench: A cross-modal hallucination benchmark for audio-visual large language models." arXiv preprint arXiv:2410.18325 (2024).
>
> [4] Radevski G, Popordanoska T, Blaschko M B, et al. DAVE: Diagnostic benchmark for Audio Visual Evaluation[J]. arXiv preprint arXiv:2503.09321, 2025.
>
> **W2&Q3: Human baseline**
>
> A: Thank you for highlighting the need for a human baseline. In response, we conducted a dedicated human evaluation designed to be **stable**, **unbiased**, and **aligned with the benchmark setting**.
>
> We invited 10 qualified annotators, including 8 **graduate students** experienced in multimodal research and 2 experts trained in **music-related analysis**. Prior to formal evaluation, we demonstrated through random sampling tests that human assessment exhibits only **minimal bias** and is feasible. We consolidated all questions requiring **Musical Knowledge** and assigned them to two music experts. The remaining questions were distributed roughly equally between **Speech** and **Sound** samples, then evenly assigned to highly qualified personnel such as graduate students to achieve persuasive results. Human responses were conducted without time constraints, achieving a final accuracy rate of **82.7%** and the results for audio classification are shown in the table below. This indicates that existing models still exhibit a **significant gap** compared to human capabilities.
>
> |Model        | Speech(%) | Sound(%) | Music(%) | <1 min(%) | 1-5 mins(%) | 5-10 mins(%) | >10 mins(%) | Avg.(%) |
> |------------------|------------------|------------------|--------------|---------------|--------------|--------------|--------------|--------------|
> | Qwen2.5-Omni-7B      | 30.70 | 25.33 | 23.07 | 41.57 | 27.41 | 25.33 | 26.72 | 29.30 |
> |**Human Baseline** | 84.12 | 74.73 | 80.27 | 78.31 | 85.71 | 80.79 | 83.21 | 82.70 |
>
> Thank you for your suggestion. We have incorporated the human benchmark results into the revised version.

---

> ### Author Response · Authors · 2025-11-20
>
> **W3: Qualitative analyses of cases**
>
> | Error Category        | Gemini2.0-Flash | Qwen3.0-Omni | Qwen2.5-Omni |
> | :-------------------- | :-------------: | :----------: | :----------: |
> | Audio Understanding   |      17.77%     |     25.6%    |     29.2%    |
> | Temporal Reasoning    |      6.15%      |     39.9%    |     28.2%    |
> | Multimodal Fusion     |      61.03%     |     23.0%    |     25.6%    |
> | Visual Understanding  |      12.03%     |     8.3%     |     9.9%     |
> | Counting              |      1.9%      |     2.0%     |     4.7%     |
> | Spatial Understanding |      1.12%      |     1.2%     |     2.4%     |
>
>
>
> Thank you for the insightful question. Qwen2.5-Omni, Qwen3-Omni-30A3B, and Gemini2.0-Flash were selected as the subjects for analysis. We used DeepSeek-V3.1 to align the models’ erroneous reasoning chains with OmniVideoBench ground-truth chains and summarized each error into 44 concise types, grouped into six high-level categories: temporal reasoning, audio comprehension, multimodal fusion, visual comprehension, counting, and spatial reasoning. **Human verification** on 50 random examples per category showed strong agreement with the automatic labels, confirming reliability.
>
> The error distributions differ between open-source and closed-source models. For **open-source models** (Qwen2.5-Omni and Qwen3-Omni-30A3B), **temporal reasoning (28–40%)**, **audio understanding (25–29%)**, and **multimodal fusion (23–26%)** dominate, accounting for ~80–90% of errors. In contrast, the **closed-source Gemini2.0-Flash** shows a distinct pattern, with **multimodal fusion (61%)** being the largest source of errors, followed by audio and visual understanding. This comparison highlights that while long-range temporal modeling, non-speech audio understanding, and cross-modal reasoning remain key challenges for closed-source MLLMs, open-source models like Gemini2.0-Flash face more severe multimodal alignment difficulties. Results are summarized in the table above.
>
> Building on the above error analysis, we further examined performance on long-video and audio-centric tasks by analyzing the distribution of error types across different tasks and audio categories, the results are shown in the table below.
> . For **long videos**, the main challenge is inadequate modeling of **long-range dependencies** and **sequential relationships**. When a target event spans multiple clips, models must integrate information across time. Additionally, visual content and corresponding audio are sometimes not perfectly synchronized, further complicating reasoning.
>
> For **music tasks**, non-speech audio **lacks symbolic semantic anchors**. Unlike speech, music and environmental sounds do not have direct text mappings, so models struggle to map continuous acoustic features such as rhythm, timbre, and intensity to semantic concepts like events or emotions. Audio source attribution also poses difficulties: as shown in the table above, models sometimes misinterpret background music as a character’s speech or incorrectly associate off-screen sounds with on-screen subjects.
>
> We also observed an intriguing pattern: although models perform relatively poorly on the **Counting and Spatial Understanding tasks**, the number of error labels falling strictly into these two categories is small. Further inspection shows that most failures originate from more fundamental **issues—fine-grained perception** and **precise temporal localization**. These tasks often require identifying small objects or their orientations at specific time points in long videos, which remains extremely challenging for current MLLMs. Errors in the earliest recognition stage propagate into incorrect counting or spatial reasoning.

---

> ### Author Response · Authors · 2025-11-20
>
> **Qwen2.5-Omni**
>
> - ERROR DISTRIBUTION BY VIDEO DURATION
> | Duration | Audio Understanding | Temporal Reasoning | Multimodal Fusion | Visual Understanding | Counting Error | Spatial Understanding |
> | :------- | :------------------ | :----------------- | :---------------- | :------------------- | :-------------- | :-------------------- |
> | <1min    | 27.7%               | 31.9%              | 27.7%             | 6.7%                 | 4.2%            | 1.7%                  |
> | 1–5min   | 23.3%               | 36.8%              | 24.5%             | 10.3%                | 3.2%            | 2.0%                  |
> | 5–10min  | 22.9%               | 45.3%              | 21.2%             | 10.0%                | 0.0%            | 0.6%                  |
> | >10min   | 29.3%               | 43.9%              | 20.0%             | 5.4%                 | 1.0%            | 0.5%                  |
>
>
>
> - ERROR DISTRIBUTION BY AUDIO TYPE
> | Audio Type | Audio Understanding | Temporal Reasoning | Multimodal Fusion | Visual Understanding | Counting Error | Spatial Understanding |
> | :--------- | :------------------ | :----------------- | :---------------- | :------------------- | :-------------- | :-------------------- |
> | Music      | 15.9%               | 47.8%              | 24.6%             | 8.7%                 | 1.4%            | 1.4%                  |
> | Sound      | 18.7%               | 52.3%              | 18.7%             | 5.6%                 | 4.7%            | 0.0%                  |
> | Speech     | 28.0%               | 36.6%              | 23.6%             | 8.8%                 | 1.6%            | 1.4%                  |
>
> **Qwen3-Omni**
> - ERROR DISTRIBUTION BY VIDEO DURATION
>
> | Duration | Audio Understanding | Temporal Reasoning | Multimodal Fusion | Visual Understanding | Counting Error | Spatial Understanding |
> | :------- | :------------------ | :----------------- | :---------------- | :------------------- | :-------------- | :-------------------- |
> | <1min    | 25.2%               | 31.1%              | 30.1%             | 7.8%                 | 5.8%            | 0.0%                  |
> | 1-5min   | 28.0%               | 26.8%              | 25.9%             | 12.1%                | 5.9%            | 1.3%                  |
> | 5-10min  | 29.7%               | 33.5%              | 24.1%             | 8.2%                 | 4.4%            | 0.0%                  |
> | >10min   | 33.8%               | 25.6%              | 25.6%             | 10.3%                | 3.1%            | 1.5%                  |
>
>
> - ERROR DISTRIBUTION BY AUDIO TYPE
>
> | Audio Type | Audio Understanding | Temporal Reasoning | Multimodal Fusion | Visual Understanding | Counting Error | Spatial Understanding |
> | :--------- | :------------------ | :----------------- | :---------------- | :------------------- | :-------------- | :-------------------- |
> | Music      | 23.4%               | 28.1%              | 31.2%             | 9.4%                 | 7.8%            | 0.0%                  |
> | Sound      | 17.8%               | 42.1%              | 21.5%             | 12.1%                | 6.5%            | 0.0%                  |
> | Speech     | 32.8%               | 26.0%              | 26.3%             | 9.7%                 | 4.0%            | 1.1%                  |
>
>
> **Gemini2.0-Flash**
> - ERROR DISTRIBUTION BY VIDEO DURATION
> | Duration | Audio Understanding | Temporal Reasoning | Multimodal Fusion | Visual Understanding | Counting Error | Spatial Understanding |
> | :------- | :------------------ | :----------------- | :---------------- | :------------------- | :-------------- | :-------------------- |
> | <1min    | 16.3%               | 10.8%              | 39.8%             | 9.6%                 | 19.3%           | 4.2%                  |
> | 1-5min   | 14.5%               | 10.4%              | 39.7%             | 9.6%                 | 18.6%           | 7.2%                  |
> | 5-10min  | 22.5%               | 16.7%              | 31.3%             | 9.3%                 | 13.2%           | 7.0%                  |
> | >10min   | 27.1%               | 17.2%              | 22.9%             | 16.0%                | 11.5%           | 5.3%                  |
> - ERROR DISTRIBUTION BY AUDIO TYPE
> | Audio Type | Audio Understanding | Temporal Reasoning | Multimodal Fusion | Visual Understanding | Counting Errorr | Spatial Understanding |
> | :--------- | :------------------ | :----------------- | :---------------- | :------------------- | :-------------- | :-------------------- |
> | Music      | 50.5%               | 9.9%               | 14.3%             | 2.2%                 | 17.6%           | 5.5%                  |
> | Sound      | 19.0%               | 12.9%              | 23.1%             | 15.6%                | 23.1%           | 6.1%                  |
> | Speech     | 16.4%               | 14.3%              | 37.7%             | 11.4%                | 13.9%           | 6.3%                  |

---

> ### Author Response · Authors · 2025-11-20
>
> **Case Study Examples**
>
> - **1. Audio Understanding Error**
>
> | Field | Content |
> |:------|:--------|
> | Question (Type) | When the man and woman were discussing ice cubes, why did they notice Superman? (Causal Reasoning) |
> | Answer (GT) | Because Superman made a sound when he stepped on the wooden floor. |
> | Model Answer | Because Superman made too much noise while eating. |
> | GT steps | - Identify the ice-cube discussion moment. |
> | | - See Superman turn and step. |
> | | - **Footstep sound** → cause of attention. |
> | Model steps | - Correctly locates the ice-cube moment. |
> | | - Correctly detects a sound at that moment. |
> | | - Interprets the sound as **eating noise**. |
> | Difference | The model misattributes the sound source, replacing the real **footstep** with an imagined **eating noise**. |
>
> ---
>
> - **2. Temporal Reasoning Error**
>
> | Field | Content |
> |:------|:--------|
> | Question (Type) | What is the fourth animal seen in the Safari Zone? (Temporal Understanding) |
> | Answer (GT) | Jackal |
> | Model Answer | Cheetah |
> | GT steps | - Use 4:50 as the Safari Zone start. |
> | | - Count animals in order → 4th is jackal. |
> | Model steps | - Correctly attempts to establish an animal sequence. |
> | | - **Incorrectly ignores the true start time.** |
> | | - **Incorrectly invents animals/scenes not present.** |
> | | - **Incorrectly concludes the 4th is cheetah.** |
> | Difference | The model breaks temporal alignment and inserts fabricated evidence, leading to an incorrect ordering. |
>
> ---
>
> - **3. Multimodal Fusion Error**
>
> | Field | Content |
> |:------|:--------|
> | Question (Type) | When a blogger presents the process of searching for a skier, which of the following is correct? (Summarization) |
> | Answer (GT) | There were 4 clips found that were not highly relevant. |
> | Model Answer | Both C and D are correct |
> | GT steps | - Confirm total clips (10) and relevant clips (6). |
> | | - Deduce 4 non-relevant clips → option B. |
> | Model steps | - Correctly tries to evaluate options through clip information. |
> | | - **Incorrectly ignores the actual 10–6 visual evidence.** |
> | | - **Incorrectly invents clip titles and durations not shown.** |
> | | - **Incorrectly concludes C and D are correct.** |
> | Difference | Model fails to fuse factual visual counts and relies on fabricated details, breaking multimodal reasoning. |
>
> ---
>
> - **4. Visual Understanding Error**
>
> | Field | Content |
> |:------|:--------|
> | Question (Type) | Where is the person with jacket-matching hair located relative to Peggy? (Spatial Understanding) |
> | Answer (GT) | Across from Peggy |
> | Model Answer | To the right of Peggy |
> | GT steps | - Identify jacket colors (orange/red). |
> | | - Find red-haired person in the scene. |
> | | - Person sits across from Peggy. |
> | Model steps | - Correctly attempts to identify the target person. |
> | | - **Incorrectly identifies the hair as blonde.** |
> | | - **Incorrectly fabricates Peggy's seating position.** |
> | | - **Incorrectly infers the person is to Peggy's right.** |
> | Difference | The model fails basic visual recognition (hair color + seating) and introduces invented details, leading to a wrong spatial judgment. |
>
> ---

---

> ### Author Response · Authors · 2025-11-20
>
> **Q1: Quantitative statistics on the efficacy of the automated filtering steps**
>
> A: Thank you for the question. Our QA filtering process consists of three stages:
>
> 1. **MLLM Filtering (Gemini 2.0 Flash):** Out of 2,500 initial QA pairs, 1,500 were retained (60%), while 1,000 pairs solvable with a single modality (vision-only or audio-only) were filtered out.
> 2. **LLM Filtering (DeepSeek-V3.1):** From the 1,500 MLLM-filtered pairs, 1,103 were retained (73.5%), with 397 text-answerable pairs removed.
> 3. **Human Verification:** The 1,103 automatically filtered pairs were reviewed by humans, resulting in 1,000 valid QA pairs (90.7%), with 103 pairs rejected due to errors or mismatched modalities.
>
> **All 1,000 final QA pairs require true audio-visual reasoning**, ensured by the combination of MLLM filtering, LLM filtering, and human validation.
>
> We have added these statistics to Section 2.4 for clarity.
>
>
> **Q2: Clarification about semantic units**
>
> A: Thank you for the question. The “semantic units” used for the semantic distance metric are **manually** defined and segmented based on meaning—such as **an object, a person, or an attribute**. This manual decomposition helps us construct balanced distractor options, ensuring that models cannot rely on textual shortcuts to answer questions without watching the video.
>
> For example, consider the question:
> “What is the relationship between the two persons?”
> with the correct answer:
> “**The person in the blue jacket is Jack, and they are colleagues.**”
>
> From this answer, we can extract several modifiable semantic units listed below. Since semantic units are manually defined, this segmentation is performed by hand.
>
> * Referring expression: “The person in the blue jacket”
> * Clothing color: “blue”
> * Clothing type: “jacket”
> * Person’s name: “Jack”
> * Relationship: “colleagues”
>
> A naive distractor-generation approach may simply alter one semantic unit at a time and each consecutive bolded segment indicates a point that differs from the correct answer,  which is a semantic unit.
>
> 1. **The speaker** is Jack, and they are colleagues.
> 2. The person in the blue jacket is **Tim**, and they are colleagues.
> 3. The person in the blue jacket is Jack, and they are **classmates**.
>
> Although each distractor differs from the correct answer by exactly one unit, the **pairwise differences among distractors are much larger**, as shown in the following table. This imbalance introduces unintended textual cues for text-only reasoning.
>
> |                    | Correct Answer | Option 1 | Option 2 | Option 3 |
> | ------------------ | -------------- | -------- | -------- | -------- |
> | **Correct Answer** | 0              | **1**    | **1**    | **1**    |
> | **Option 1**       | **1**          | 0        | 2        | 2        |
> | **Option 2**       | **1**          | 2        | 0        | 2        |
> | **Option 3**       | **1**          | 2        | 2        | 0        |
>
> To avoid this, we require that the semantic differences **cannot form a fully unbalanced pattern**—that is, the correct option cannot differ from all distractors by a uniform value *a* while the distractors differ from one another by a different uniform value *b*, where ***a*** ≠ ***b***
> . This constraint prevents models from locating the correct answer by exploiting this structural asymmetry.
>
> Thus, we modify the distractors to ensure **balanced pairwise semantic distances**:
>
> 1. **The speaker** is **Tim**, and they are colleagues.
> 2. The person in the blue jacket is **Tim**, and they are **classmates**.
> 3. The speaker is Jack, and they are **classmates**.
>
> With these revised distractors, the pairwise semantic distances become more heterogeneous and no longer exhibit the previous uniform imbalance. For example, Option 1 differs from Option 3 by 3 semantic units, while Option 2 differs from them by 2, as shown below. In this way, the model must correctly understand the video content to answer the question, rather than simply guessing the answer from the text.
>
> |                    | Correct Answer | Option 1 | Option 2 | Option 3 |
> | ------------------ | -------------- | -------- | -------- | -------- |
> | **Correct Answer** | 0              | 2        | 2        | 1        |
> | **Option 1**       | 2              | 0        | 2        | 3        |
> | **Option 2**       | 2              | 2        | 0        | 2        |
> | **Option 3**       | 1              | 3        | 2        | 0        |

---

> ### Author Response · Authors · 2025-11-23
> **Polite follow-up: request for reviewers’ feedback on our responses!**
>
> Hi, we sincerely thank you very much for these **constructive comments and evaluation** of our manuscript. We would like to kindly ask you to **take a look at our responses** and **reevaluate our work** based on our clarifications. Please let us know **whether our response addresses your concerns** or **whether there is any further detail we can provide** to help address these concerns.
>
> Thank you again for dedicating your time to reviewing our paper.

---

> > ### Author Response · Authors · 2025-11-25
> >
> > Dear Reviewer rRon,
> >
> > I hope this message finds you well. We are writing to gently follow up regarding our rebuttal. We understand that this is a busy period, but if you could spare a moment to review our responses and share any updated evaluation, we would be truly grateful.
> >
> > Please let us know if any additional clarification or information from our side would be helpful. We sincerely appreciate your time and effort in reviewing our work.
> >
> > Warm regards,
> >
> > Authors of OmniVideoBench

---

> ### Author Response · Authors · 2025-11-27
>
> Dear Reviewer,
>
> We hope that the clarifications and additional experiments provided in the rebuttal have sufficiently addressed your concerns. We have made our best efforts to respond to all questions, and sincerely hope our responses have clarified the issues raised. If you have any further questions or would like additional details, please feel free to let us know
>
> Sincerely,
>
> Authors

---

> ### Author Response · Authors · 2025-11-28
>
> Dear Reviewer rRon,
>
> We understand that chasing down your reply is not our job and we do not intend to add any pressure on your busy schedule. However, as we are getting closer to the end of the discussion phase, we would really appreciate it if you could be so kind to let us know if we have properly addressed your comments and questions in the rebuttal and if anything can be further clarified.
>
> Many thanks in advance!
>
> Authors

---

### Official Review · Reviewer_adN5 · 2025-11-02

**Soundness:** 3
**Presentation:** 2
**Contribution:** 3
**Rating:** 4
**Confidence:** 4

**Summary:**

The paper presents OmniVideoBench, a large-scale, carefully curated benchmark evaluating synergistic audio-visual reasoning in MLLMs. OmniVideoBench consists of 1,000 manually verified QA pairs with explicit step-by-step reasoning traces, based on 628 diverse long-form videos across 8 major categories and 68 subcategories. The benchmark emphasizes logical consistency, modality complementarity, and covers 13 question types relevant to real-world video understanding. The authors systematically filter and annotate the data to ensure questions demand audio-visual integration. Baseline experiments on both open- and closed-source MLLMs reveal a substantial gap to human-level reasoning, particularly regarding music, long videos, and abstract audio understanding.

**Strengths:**

1. The benchmark features 1,000 high-quality, manually verified QA examples, each annotated with step-by-step reasoning chains with human proofreading. This explicit annotation supports analysis of both model answers and reasoning processes.
2. Long Videos span 8 major categories and 68 subcategories, ensuring comprehensive coverage and evaluation of a wide range of real-world scenarios.
3. The authors conducted extensive and comprehensive evaluations and experiments.

**Weaknesses:**

1. Missing Comparisons with Key Recent Audio-Visual Benchmarks: Several highly relevant, recently released benchmarks should be compared and discussed [1,2,3].
2. It will be great to see the error analysis on the properties of the questions/items themselves—e.g., what makes some reasoning chains or audio-visual interactions particularly difficult?
3. More qualitative examples and error cases could be included in the appendix to provide deeper insights into model behavior and help readers better understand the paper.








[1] Sung-Bin, Kim, et al. "Avhbench: A cross-modal hallucination benchmark for audio-visual large language models." arXiv preprint arXiv:2410.18325 (2024).
[2]Chowdhury, Sanjoy, et al. "Avtrustbench: Assessing and enhancing reliability and robustness in audio-visual llms." arXiv preprint arXiv:2501.02135 (2025).
[3[ Sakshi, S., et al. "Mmau: A massive multi-task audio understanding and reasoning benchmark." arXiv preprint arXiv:2410.19168 (2024).

**Questions:**

Please check the Weaknesses.

---

> ### Author Response · Authors · 2025-11-20
>
> **Q1: Missing Comparisons with Key Recent Audio-Visual Benchmarks**
>
> A1:Thank you for your suggestion! **We agree that the benchmarks you mentioned are highly relevant**, and we have added **discussions and comparisons** related to them in the **latest version of the paper**.
> Compared to pure audio benchmarks like **MMAU** [1], OmniVideoBench thoroughly examines MLLMs' ability to **integrate audio–visual information**, moving beyond **single-modality limitations**.
>
> Unlike benchmarks such as **AVHBench** [2] and **AVTRUSTBENCH** [3], which focus on evaluating shorter videos (under 1 minute), OmniVideoBench extends the maximum video duration to **1955 seconds** with an average of **384 seconds**, filling an important gap in long-video evaluation and more comprehensively challenging models' ability to capture extended contextual information.
>
> For instance, Qwen2.5-Omni-7B achieves **74.7%** on the AVHBench Judgment task but only **29.3%** on OmniVideoBench, further demonstrating the added difficulty and diagnostic value of our benchmark.
> **More details** are now available in Section 2.6 DATASET COMPARISON of the updated paper.
>
> | Benchmark                 | Modality | Qwen2.5-Omni | Multiple Domains | Video Type | Audio Type     | Video Duration | Answer Type |
> |---------------------------|----------|--------------|------------------|------------|----------------|----------------|-------------|
> | AVQA                      | V+A      | /            | ✗                | R          | So             | 10             | MC          |
> | Music-AVQA                | V+A      | /            | ✗                | R+S        | Mu             | 60             | CLS         |
> | AVTRUSTBENCH              | V+A      | /            | ✓                | R+S        | Sp+So+Mu       | 10/60          | MC          |
> | MMAU                      | A        | 71.0         | ✓                | /          | Sp+So+Mu       | /              | MC          |
> | DAVE                      | V+A      | 31.0         | ✓                | R+S        | So             | ≤60            | MC          |
> | AV-Odyssey                | I+A      | /            | ✓                | R          | Sp+So+Mu       | /              | MC          |
> | AVHBench (Judgement)      | V+A      | 74.7         | ✓                | R+S        | So             | 10             | CLS         |
> | OmniBench                 | I+A      | 56.1         | ✓                | R          | Sp+So+Mu       | /              | MC          |
> | Daily-Omni                | V+A      | 47.5         | ✗                | R          | Sp+So+Mu       | 30/60          | MC          |
> | WorldSense                | V+A      | 48.3         | ✓                | R          | Sp+So+Mu       | 15–656         | MC          |
> | **OmniVideoBench (Ours)** | V+A      | **29.3**     | ✓                | R          | Sp+So+Mu       | **4–1955**     | MC          |
>
>
> [1] Sakshi, S., et al. "Mmau: A massive multi-task audio understanding and reasoning benchmark." arXiv:2410.19168 (2024).
>
> [2] Sung-Bin, Kim, et al. "Avhbench: A cross-modal hallucination benchmark for audio-visual large language models." arXiv:2410.18325 (2024).
>
> [3] Chowdhury, Sanjoy, et al. "Avtrustbench: Assessing and enhancing reliability and robustness in audio-visual llms." arXiv:2501.02135 (2025).

---

> ### Author Response · Authors · 2025-11-20
>
> **Q2: Lack of error analysis**
>
> A2: Thank you for this excellent suggestion. We performed a dedicated error analysis to understand which types of reasoning chains and audio–visual interactions are particularly challenging for current MLLMs. You can find our error statistics methodology, conclusions, and some examples in **Appendix D** of **the latest version of the paper**.
>
> Using step-by-step reasoning chains generated by Qwen2.5-Omni and Qwen3-Omni, we aligned incorrect chains with our ground-truth chains and summarized the underlying causes into interpretable categories. Across 44 refined error types, three major categories dominate: **temporal reasoning error (39.9%)**, **audio understanding error (25.6%)**, and **multimodal fusion error (23.0%)**, together accounting for nearly **89%** of all failures. In contrast, the closed-source Gemini2.0-Flash shows a distinct pattern, with multimodal fusion error (61%) being the largest source of errors, followed by audio and visual understanding. This comparison highlights that while long-range temporal modeling, non-speech audio understanding, and cross-modal reasoning remain key challenges for closed-source MLLMs, open-source models like Gemini2.0-Flash face more severe multimodal alignment difficulties. Results are summarized in the table.
>
> | Error Category        | Gemini2.0-Flash | Qwen3.0-Omni | Qwen2.5-Omni |
> | :-------------------- | :-------------: | :----------: | :----------: |
> | Audio Understanding   |      17.77%     |     25.6%    |     29.2%    |
> | Temporal Reasoning    |      6.15%      |     39.9%    |     28.2%    |
> | Multimodal Fusion     |      61.03%     |     23.0%    |     25.6%    |
> | Visual Understanding  |      12.03%     |     8.3%     |     9.9%     |
> | Counting              |      1.9%      |     2.0%     |     4.7%     |
> | Spatial Understanding |      1.12%      |     1.2%     |     2.4%     |
>
> These patterns suggest that the most difficult properties of OmniVideoBench questions arise from (1) **long-range temporal dependencies**, where models struggle to integrate visual and auditory evidence across far-apart segments; (2) **non-speech audio understanding**, especially music and environmental sounds that lack symbolic representations and require direct acoustic inference; and (3) **cross-modal alignment**, where models often rely on a single modality instead of validating audio and vision jointly, or fail to synchronize them temporally.
>
> Notably, although tasks such as Counting and Spatial Understanding yield low accuracy, their error chains typically trace back to upstream issues in **fine-grained perception** or **precise temporal localization** rather than counting or spatial reasoning alone. This highlights how early-stage perception errors propagate through the reasoning chain in long, complex videos.
> Overall, our analysis shows that the hardest items in OmniVideoBench are those requiring **extended temporal integration**, **interpreting non-speech audio**, or **tight audio–visual coherence**, which are precisely the areas where current MLLMs remain limited. These findings align directly with the design goals of OmniVideoBench and confirm that the benchmark exposes fundamental weaknesses in state-of-the-art audio–visual reasoning models.

---

> ### Author Response · Authors · 2025-11-20
>
> **Q3: Qualitative examples and error cases**
>
> A3: Thank you for your nice comments and suggestions. We have added more details regarding the error analysis in the appendix of the latest version of the paper. You can find our analysis results and some reference examples in **Appendix D**.
>
> To delve deeper into the reasons behind the model's successes and failures on the OmniVideoBench test set, we further guided Qwen2.5-Omni and Qwen3-Omni to generate **step-by-step reasoning chains** through prompt outputs as shown in Appendix C. We compared the results obtained from the reasoning chain prompts with those from the standard “direct response letter” evaluation. The overall accuracy difference consistently remained below 1%, indicating that reasoning-based generation **does not significantly alter model predictions** and can thus be **reliably used for reasoning analysis**.
>
> After collecting reasoning chains, we employed the DeepSeek-V3.1 model to align erroneous reasoning chains with the ground-truth chains annotated in OmniVideoBench. The model was tasked with summarizing the core issue of each prediction error using concise labels. This process generated **44 refined error types**, clustered into **six major categories**: **audio comprehension error**, **temporal reasoning error**, **multimodal fusion error**, **visual comprehension error**, **counting error**, and **spatial comprehension error**.
>
> To validate the reliability of this automated annotation pipeline, we randomly selected 50 examples from each category for independent **human review**, and the human annotations showed high consistency with the automated results, demonstrating the stability and reliability of this annotation workflow.
>
> You can find examples of error analysis in the table below. For more detailed and intuitive case studies, please refer to **Appendix D** of the latest version of the paper.

---

> ### Author Response · Authors · 2025-11-21
> **More details about error analysis**
>
> **Qwen2.5-Omni**
>
> - ERROR DISTRIBUTION BY VIDEO DURATION
> | Duration | Audio Understanding | Temporal Reasoning | Multimodal Fusion | Visual Understanding | Counting Error | Spatial Understanding |
> | :------- | :------------------ | :----------------- | :---------------- | :------------------- | :-------------- | :-------------------- |
> | <1min    | 27.7%               | 31.9%              | 27.7%             | 6.7%                 | 4.2%            | 1.7%                  |
> | 1–5min   | 23.3%               | 36.8%              | 24.5%             | 10.3%                | 3.2%            | 2.0%                  |
> | 5–10min  | 22.9%               | 45.3%              | 21.2%             | 10.0%                | 0.0%            | 0.6%                  |
> | >10min   | 29.3%               | 43.9%              | 20.0%             | 5.4%                 | 1.0%            | 0.5%                  |
>
>
>
> - ERROR DISTRIBUTION BY AUDIO TYPE
> | Audio Type | Audio Understanding | Temporal Reasoning | Multimodal Fusion | Visual Understanding | Counting Error | Spatial Understanding |
> | :--------- | :------------------ | :----------------- | :---------------- | :------------------- | :-------------- | :-------------------- |
> | Music      | 15.9%               | 47.8%              | 24.6%             | 8.7%                 | 1.4%            | 1.4%                  |
> | Sound      | 18.7%               | 52.3%              | 18.7%             | 5.6%                 | 4.7%            | 0.0%                  |
> | Speech     | 28.0%               | 36.6%              | 23.6%             | 8.8%                 | 1.6%            | 1.4%                  |
>
> **Qwen3-Omni**
> - ERROR DISTRIBUTION BY VIDEO DURATION
>
> | Duration | Audio Understanding | Temporal Reasoning | Multimodal Fusion | Visual Understanding | Counting Error | Spatial Understanding |
> | :------- | :------------------ | :----------------- | :---------------- | :------------------- | :-------------- | :-------------------- |
> | <1min    | 25.2%               | 31.1%              | 30.1%             | 7.8%                 | 5.8%            | 0.0%                  |
> | 1-5min   | 28.0%               | 26.8%              | 25.9%             | 12.1%                | 5.9%            | 1.3%                  |
> | 5-10min  | 29.7%               | 33.5%              | 24.1%             | 8.2%                 | 4.4%            | 0.0%                  |
> | >10min   | 33.8%               | 25.6%              | 25.6%             | 10.3%                | 3.1%            | 1.5%                  |
>
>
> - ERROR DISTRIBUTION BY AUDIO TYPE
>
> | Audio Type | Audio Understanding | Temporal Reasoning | Multimodal Fusion | Visual Understanding | Counting Error | Spatial Understanding |
> | :--------- | :------------------ | :----------------- | :---------------- | :------------------- | :-------------- | :-------------------- |
> | Music      | 23.4%               | 28.1%              | 31.2%             | 9.4%                 | 7.8%            | 0.0%                  |
> | Sound      | 17.8%               | 42.1%              | 21.5%             | 12.1%                | 6.5%            | 0.0%                  |
> | Speech     | 32.8%               | 26.0%              | 26.3%             | 9.7%                 | 4.0%            | 1.1%                  |
>
>
> **Gemini2.0-Flash**
> - ERROR DISTRIBUTION BY VIDEO DURATION
> | Duration | Audio Understanding | Temporal Reasoning | Multimodal Fusion | Visual Understanding | Counting Error | Spatial Understanding |
> | :------- | :------------------ | :----------------- | :---------------- | :------------------- | :-------------- | :-------------------- |
> | <1min    | 16.3%               | 10.8%              | 39.8%             | 9.6%                 | 19.3%           | 4.2%                  |
> | 1-5min   | 14.5%               | 10.4%              | 39.7%             | 9.6%                 | 18.6%           | 7.2%                  |
> | 5-10min  | 22.5%               | 16.7%              | 31.3%             | 9.3%                 | 13.2%           | 7.0%                  |
> | >10min   | 27.1%               | 17.2%              | 22.9%             | 16.0%                | 11.5%           | 5.3%                  |
> - ERROR DISTRIBUTION BY AUDIO TYPE
> | Audio Type | Audio Understanding | Temporal Reasoning | Multimodal Fusion | Visual Understanding | Counting Error | Spatial Understanding |
> | :--------- | :------------------ | :----------------- | :---------------- | :------------------- | :-------------- | :-------------------- |
> | Music      | 50.5%               | 9.9%               | 14.3%             | 2.2%                 | 17.6%           | 5.5%                  |
> | Sound      | 19.0%               | 12.9%              | 23.1%             | 15.6%                | 23.1%           | 6.1%                  |
> | Speech     | 16.4%               | 14.3%              | 37.7%             | 11.4%                | 13.9%           | 6.3%                  |

---

> ### Author Response · Authors · 2025-11-21
> **Case Study Examples**
>
> **Case Study Examples**
>
> - **1. Audio Understanding Error**
>
> | Field | Content |
> |:------|:--------|
> | Question (Type) | When the man and woman were discussing ice cubes, why did they notice Superman? (Causal Reasoning) |
> | Answer (GT) | Because Superman made a sound when he stepped on the wooden floor. |
> | Model Answer | Because Superman made too much noise while eating. |
> | GT steps | - Identify the ice-cube discussion moment. |
> | | - See Superman turn and step. |
> | | - **Footstep sound** → cause of attention. |
> | Model steps | - Correctly locates the ice-cube moment. |
> | | - Correctly detects a sound at that moment. |
> | | - Interprets the sound as **eating noise**. |
> | Difference | The model misattributes the sound source, replacing the real **footstep** with an imagined **eating noise**. |
>
> ---
>
> - **2. Temporal Reasoning Error**
>
> | Field | Content |
> |:------|:--------|
> | Question (Type) | What is the fourth animal seen in the Safari Zone? (Temporal Understanding) |
> | Answer (GT) | Jackal |
> | Model Answer | Cheetah |
> | GT steps | - Use 4:50 as the Safari Zone start. |
> | | - Count animals in order → 4th is jackal. |
> | Model steps | - Correctly attempts to establish an animal sequence. |
> | | - **Incorrectly ignores the true start time.** |
> | | - **Incorrectly invents animals/scenes not present.** |
> | | - **Incorrectly concludes the 4th is cheetah.** |
> | Difference | The model breaks temporal alignment and inserts fabricated evidence, leading to an incorrect ordering. |
>
> ---
>
> - **3. Multimodal Fusion Error**
>
> | Field | Content |
> |:------|:--------|
> | Question (Type) | When a blogger presents the process of searching for a skier, which of the following is correct? (Summarization) |
> | Answer (GT) | There were 4 clips found that were not highly relevant. |
> | Model Answer | Both C and D are correct |
> | GT steps | - Confirm total clips (10) and relevant clips (6). |
> | | - Deduce 4 non-relevant clips → option B. |
> | Model steps | - Correctly tries to evaluate options through clip information. |
> | | - **Incorrectly ignores the actual 10–6 visual evidence.** |
> | | - **Incorrectly invents clip titles and durations not shown.** |
> | | - **Incorrectly concludes C and D are correct.** |
> | Difference | Model fails to fuse factual visual counts and relies on fabricated details, breaking multimodal reasoning. |
>
> ---
>
> - **4. Visual Understanding Error**
>
> | Field | Content |
> |:------|:--------|
> | Question (Type) | Where is the person with jacket-matching hair located relative to Peggy? (Spatial Understanding) |
> | Answer (GT) | Across from Peggy |
> | Model Answer | To the right of Peggy |
> | GT steps | - Identify jacket colors (orange/red). |
> | | - Find red-haired person in the scene. |
> | | - Person sits across from Peggy. |
> | Model steps | - Correctly attempts to identify the target person. |
> | | - **Incorrectly identifies the hair as blonde.** |
> | | - **Incorrectly fabricates Peggy's seating position.** |
> | | - **Incorrectly infers the person is to Peggy's right.** |
> | Difference | The model fails basic visual recognition (hair color + seating) and introduces invented details, leading to a wrong spatial judgment. |
>
> ---

---

> ### Author Response · Authors · 2025-11-23
> **Polite follow-up: request for reviewers’ feedback on our responses!**
>
> Hi, we sincerely thank you very much for these **constructive comments and evaluation** of our manuscript. We would like to kindly ask you to **take a look at our responses** and **reevaluate our work** based on our clarifications. Please let us know **whether our response addresses your concerns** or **whether there is any further detail we can provide** to help address these concerns.
>
> Thank you again for dedicating your time to reviewing our paper.

---

> > ### Author Response · Authors · 2025-11-25
> >
> > Dear Reviewer adN5,
> >
> > I hope this message finds you well. We are writing to gently follow up regarding our rebuttal. We understand that this is a busy period, but if you could spare a moment to review our responses and share any updated evaluation, we would be truly grateful.
> >
> > Please let us know if any additional clarification or information from our side would be helpful. We sincerely appreciate your time and effort in reviewing our work.
> >
> > Warm regards,
> >
> > Authors of OmniVideoBench

---

> > > ### Comment · Reviewer_adN5 · 2025-11-25
> > >
> > > I appreciate the authors' hard work during the rebuttal. All my concerns have been addressed, and I increased my score accordingly.

---

> > > > ### Author Response · Authors · 2025-11-26
> > > >
> > > > Thanks for your quick feedback. We will carefully address your concerns in our new version.

---

### Author Response · Authors · 2025-11-29

Thanks for handling and reviewing our submitted manuscript: "OmniVideoBench: Towards Audio-Visual Understanding Evaluation for Omni MLLMs". We sincerely thank the reviewers for their insightful and constructive comments. We are encouraged that the reviewers recognized the innovation, challenge, comprehensiveness, and high quality of OmniVideoBench. By addressing these points, we believe the quality, clarity, and completeness of our benchmark have been significantly improved. The major responses are summarized as follows:

1. **Benchmark comparisons across existing audio-visual datasets** (reviewer adN5 Q1, reviewer rRon W1)
   We added comparisons with MMAU, AVHBench, DAVE, AVQA, and WorldSense, highlighting OmniVideoBench’s long videos, multi-task coverage, and multimodal reasoning challenges.

2. **Error analysis by reasoning difficulty, modality, and video/audio properties** (reviewer adN5 Q2, reviewer rRon W3)
   Errors were categorized into 6 major types and 44 subtypes. We compared models, analyzed errors by video length and audio type, and presented representative cases.

3. **Human baseline across tasks and audio modalities** (reviewer iHcc W3, reviewer rRon W2)
   Inviting 10 annotators (including 2 music experts), we measured 82.7% overall accuracy, 84.12% on speech, and 80.27% on music tasks, providing a clear reference for models.

4. **Audio type distribution** (reviewer rRon W1, reviewer iHcc W2)
   We defined audio types as those needed to solve the question. Analysis and stability experiments show the distribution is realistic and reasonable.

5. **Multi-stage filtering and quality control of questions** (reviewer rRon Q1)
   Our filtering retained 1,000 high-quality audio-visual questions: MLLM 60%, LLM 73.5%, human verification 90.7%.

6. **Semantic units and distractor design** (reviewer rRon Q2)
   Semantic units were manually segmented meaningful elements used to design distractors, ensuring models rely on video content.

7. **Task coverage and taxonomy** (reviewer iHcc W1, reviewer xiBm Q1)
   We designed 13 task types from prior benchmarks. The “other” category ≤0.03%, and stability tests confirm task distribution is sufficient.

8. **Video quantity, domain diversity, and generalization** (reviewer xiBm Q3)
   We curated 628 videos across 8 domains and 68 subcategories from 1,800. Stability tests confirm generalization; sample data and evaluation code are provided in an anonymous repository.

9. **Comparison with WorldSense and long-video design** (reviewer iHcc Q2&Q3)
   OmniVideoBench is more challenging, with longer videos, higher task difficulty, and stronger multimodal fusion, highlighting the value of long-video reasoning.

10. **Directions for improving MLLM performance on long-form reasoning** (reviewer xiBm Q2)
    We suggest three directions: (i) high-quality audio-visual training data, (ii) long-form reasoning optimization, and (iii) cross-modal alignment improvements.

We again sincerely thank the reviewers for their constructive comments. Addressing these points has improved the clarity, comprehensiveness, and rigor of OmniVideoBench.

---

### Meta-Review · Area_Chair_oHgn · 2025-12-23

**Summary:**

The recommendation of acceptance is based on the fact that most major concerns of the reviewers have been well addressed with extensive rebuttal experiments and further clarifications. Two reviewers that were initially negative with a score of 4 have explicitly mentioned will change the score.The AC agrees with the reviewers and thinks most of the key concerns and questions are well addressed in rebuttal. I would encourage the authors to incorporate all promised changes and new results in camera ready. See detailed decision rationale below.

**Reviewer Concerns:**

See details below.

**Reviewer Scores:**

***Reviewer adN5 likely to change the score from 4 to 6 as all concerns have been well addressed and the reviewer explicitly mentioned "I increased my score accordingly."***

1. Missing Comparisons with Key Recent Audio-Visual Benchmarks

Well addressed

2. error analysis on the properties of the questions/items themselves

Addressed

3. More qualitative examples and error cases

Addressed

***Reviewer rRon likely to raise the score from 4 to 6 as the authors have done an extensive job to address the concerns. Most are well addressed. ***

1. does not provide statistical results comparing it with other relevant datasets

Well addressed

2. does not report human baseline accuracy or response variability for the main test set.

Addressed

3. lacks a deeper set of qualitative analyses of successful versus failure cases

Mostly addressed

4. provide quantitative statistics on the efficacy of the automated filtering steps in Section 2.4?

Addressed

5. clarify how "semantic units" are operationalized for the semantic distance metric

Addressed

6. human benchmark results (e.g., accuracy, agreement rates) on OmniVideoBench?

Mostly addressed

***Reviewer iHcc likely to increase the score from 4 to 6 as most major concerns are well-addressed, and the reviewer mentioned "have adjusted the score accordingly."***

1. Limited Per-Task Coverage

Mostly addressed

2. Unbalanced Audio Category Distribution

Mostly addressed

3. Lack of Human Baseline

Addressed

4. Some other clarification questions

Addressed


***Reviewer xiBm likely to maintain the positive rating of 6 as the main concern is addressed, and the reviewer explicitly mentioned "keep my original rating". ***

1. Question Type

Well addressed

2. Method Insight

Mostly addressed

3. Small Size

Mostly addressed

---

### Decision · Program_Chairs · 2026-01-26

Accept (Poster)